# Ambient-pressure alkoxycarbonylation for sustainable synthesis of ester

Bin Zhang [1,2,8], Haiyang Yuan[3,8], Ye Liu[1,4,8], Zijie Deng[1,5], Mark Douthwaite [2] ✉,
Nicholas F. Dummer [2], Richard J. Lewis [2], Xingwu Liu[6], Sen Luan[1,5],
Minghua Dong[1,5], Tianjiao Wang[1,5], Qingling Xu[5] ✉, Zhijuan Zhao[7],
Huizhen Liu [1,5] ✉, Buxing Han [1,5] & Graham J. Hutchings [2]

Alkoxycarbonylation reactions are common in the chemical industry, yet process sustainability is limited by the inefficient utilization of CO. In this study, we address this issue and demonstrate that significant improvements can be achieved by adopting a heterogeneously catalyzed process, using a Ru/NbO$_x$ catalyst. The Ru/NbO$_x$ catalyst enables the direct synthesis of methyl propionate, a key industrial commodity, with over 98% selectivity from CO, ethylene and methanol, without any ligands or acid/base promoters. Under ambient CO pressure, a high CO utilization efficiency (336 mmol$_{ester}$mol$_{CO}^{-1}$h$^{-1}$) is achieved. Mechanistic investigations reveal that CO undergoes a methoxycarbonyl (COOCH$_3$) intermediate pathway, attacking the terminal carbon atom of alkene and yielding linear esters. The origins of prevailing linear regioselectivity in esters are revealed. The infrared spectroscopic feature of the key COOCH$_3$ species is observed at 1750 cm$^{-1}$ (C=O vibration) both experimentally and computationally. The broad substrate applicability of Ru/NbO$_x$ catalyst for ester production is demonstrated. This process offers a sustainable and efficient approach with high CO utilization and atom economy for the synthesis of esters.

Carbonylation processes involve the incorporation of CO molecules into organic compounds[1]. The alkoxycarbonylation of alkenes is among the most common carbonylation reactions employed industrially and is used to produce millions of tons of esters and acids annually[1–3]. The global annual production of acetic acid was 9.1 million tons in 2019[4]; approximately 85% of which is produced from methanol carbonylation[5]. Another noteworthy application is the industrial production of methyl propionate (MP), a crucial intermediate for the synthesis of methyl methacrylate polymer and plastics (produced on a 4-million-ton scale annually)[6]. The state-of-the-art commercial process to produce MP (the Lucite Alpha process), involves the toluenesulfonic acid-promoted hydromethoxycarbonylation of ethylene. This process employs a molecular Pd catalyst ligated with phosphine ligand and requires high pressures of CO[7,8]. Thus, developing a sustainable (and

[1]Beijing National Laboratory for Molecular Sciences, CAS Laboratory of Colloid and Interface and Thermodynamics, Center for Carbon Neutral Chemistry, Institute of Chemistry, Chinese Academy of Sciences, 100190 Beijing, China. [2]Max Planck–Cardiff Centre on the Fundamentals of Heterogeneous Catalysis FUNCAT, Cardiff Catalysis Institute, School of Chemistry, Cardiff University, Cardiff CF24 4HQ, UK. [3]Key Laboratory for Ultrafine Materials of Ministry of Education, Shanghai Engineering Research Center of Hierarchical Nanomaterials, School of Materials Science and Engineering, East China University of Science and Technology, 130 Meilong Road, 200237 Shanghai, China. [4]Laboratory of Living Materials at the State Key Laboratory of Advanced Technology for Materials Synthesis and Processing, Wuhan University of Technology, 430070 Wuhan, Hubei, China. [5]School of Chemical Sciences, University of Chinese Academy of Sciences, 101408 Beijing, China. [6]National Energy Center for Coal to Liquids, Synfuels China Co., Ltd, Huairou District, 101400 Beijing, China. [7]Institute of Chemistry, Chinese Academy of Sciences, 100190 Beijing, China. [8]These authors contributed equally: Bin Zhang, Haiyang Yuan, Ye Liu. ✉e-mail: douthwaitejm@cardiff.ac.uk; xuqingling@ucas.ac.cn; liuhz@iccas.ac.cn

efficient) carbonylation process, that can be used to produce MP is of utmost importance (Fig. 1a).

The two primary challenges associated with alkoxycarbonylation reactions are linked with the need to improve the regioselectivity of the desired product and increase CO utilization[9–12]. Modifying the ligand structure of molecular Pd catalysts has proven to be an effective means of manipulating the regioselectivity in these classes of reactions[13–17]. However, high pressures of CO (often >40 bar) are typically necessary to inhibit the decarbonylation of the acyl metal intermediates[18–20], and consequentially low rates of CO conversion are often observed (<1%, Fig. 1b). Furthermore, acid promoters are typically required to maintain the catalytic activity of the Pd complexes, assisting with the formation (and stabilization) of Pd hydride species, but lead to concerns associated with reactor corrosion and product separation[21–24]. On the contrary, heterogeneous solid catalysts can accomplish the process without acid and can be easily separated from the post-reaction mixture, resulting in reduced overhead costs and waste generation[25]. Most recently, solid catalysts have been used in carbonylation reactions like hydroformylation[26–29] and aminoacylation[30–32]. For alkoxycarbonylation[33], ceria-supported ruthenium catalysts have proven to be highly effective in the absence of acid promoters, but as a consequence, they are limited by CO utilization (up to 45 $mmol_{ester}mol_{CO}^{-1}h^{-1}$ reported, Fig. 1b)[34,35]. Thus, developing a heterogeneous process that can deliver effective alkoxycarbonylation, especially under atmospheric-pressure CO, is highly desirable, as it would improve CO utilization and overall process efficiency. Furthermore, controlling regioselectivity in heterogeneous catalytic systems is still considered uncharted territory, yet holds immense value[36,37].

In this study, we explored the heterogeneous alkoxycarbonylation of alkenes with CO and alcohols under atmospheric pressure. MP was successfully synthesized with over 98% selectivity over a Ru/NbO$_x$ catalyst, in the absence of an acid promoter, with a CO utilization efficiency (336 $mmol_{MP}mol_{CO}^{-1}h^{-1}$). The efficiency of CO utilization was three orders of magnitudes higher than the homogenous molecular catalysts reported (Fig. 1a, b, Supplementary Table 1). Yet there was no issue in regioselectivity when ethylene was used as the reactant. Subsequently, styrene as a model molecule was employed to investigate regioselective hydromethoxycarbonylation (Fig. 1c). The reaction mechanism involving multiple reactive species over Ru/NbO$_x$ was examined, revealing the origins of the predominance of linear-selective esters in a heterogeneous system. Additionally, we explored the control of regioselectivity toward linear (anti-Markovnikov) and branched (Markovnikov) esters and examined a broad substrate scope, including various alkenes and alcohols.

## Results
### Catalytic methoxycarbonylation reaction
Under the reaction conditions investigated, there exist three competing pathways for the catalytic reactions involving styrene, CO and methanol: hydrogenation, methoxycarbonylation and methoxylation. To ensure optimal carbonylation efficiency, there is clearly a need to identify a suitable catalyst that can selectively drive styrene toward carbonylation, while simultaneously inhibiting the undesirable hydrogenation and methoxylation pathways (Fig. 1c). Our initial investigations indicated that ruthenium is a highly effective active metal for this application (Fig. 1d, Supplementary Table 2). The commercially available carbon catalysts (Pt/C, Pd/C, Ir/C) showed major methoxylation products due to the strong acidity of activated carbon support, similar to acidic support catalysts (Ru/Hβ, Ru/WZrO$_x$). The Ru/C catalyst exhibited a dominant hydrogenation pathway due to the strong metallicity of large-size Ru nanoparticles. The ceria-based Ru catalysts (Ru/CeO$_2$, Ru/CeO$_x$) showed distinct carbonylation selectivity (80–90%) despite poor conversion, due to their ability to form oxygen vacancy defects (O$_v$) on the surface. These oxygen defects can serve as

active sites for anchoring and dispersing the Ru metal, as well as for methanol dissociation at the Ce-O$_v$ site[34,35]. Our recent work has found that NbO$_x$-based catalysts are effective at activating and dissociating the C-O bonds[38–40], due to the abundant oxygen defects created through hydrogen treatment, despite Nb$_2$O$_5$ being typically less reducible than CeO$_2$. The defective surface, along with its intrinsic acidic properties, makes NbO$_x$-based catalysts potentially catalytically active for promoting the hydromethoxycarbonylation reaction.

The synthesized Ru/NbO$_x$ catalyst exhibited a remarkable selectivity (up to 90%) to esters at 13% conversion, under 1 MPa of CO. Rather surprisingly, when the reaction pressure was reduced to 0.1 MPa a significant increase in conversion (55%) was observed, at no detriment to the ester selectivity (90%) (Fig. 1d). Encouraged by the good performance of the Ru/NbO$_x$ catalyst, additional experiments to optimize the reaction conditions were conducted (Supplementary Tables 4–10). These studies revealed that higher temperatures and CO pressures were unfavorable for the reaction. The optimal temperature was found to be 170 °C, as higher temperatures promoted the competitive hydrogenation pathway. At 200 °C, the reaction showed a 40% decrease in the selectivity of carbonylation but doubled its selectivity of hydrogenation (up to 56%), in comparison to the reaction at 170 °C (Supplementary Table 4). In the absence of CO, hydrogenation became the primary pathway, with a comparatively low conversion of approximately 3%. Styrene conversion is optimal near ambient pressures of CO (0.06–0.1 MPa) and decreases rapidly with increasing pressure, as does the CO utilization, which was speculated to be limited by styrene concentration. Then we increased the styrene concentration by tenfold at a fixed CO pressure, and achieved a CO utilization of 62.4% (Supplementary Table 1). Interestingly, this did not appear to have a significant impact on the reaction selectivity, as the product distribution remained virtually unchanged (Fig. 1e). This decrease in activity (as pressure is increased) is likely to be attributed to an increased surface coverage of CO on the Ru (namely the occupation of more Ru surface), thus inhibiting the adsorption of other reactive species (e.g., styrene, CH$_3$O* and H*). The progress of the reaction over time was subsequently investigated at 0.1 MPa CO (Fig. 1f). A high regioselectivity (ca. 75%) to linear esters was observed throughout the reaction and, thus, did not appear to be influenced by the conversion. After 16 h of reaction, the styrene conversion reached >95%, and the ester yield reached 72%, with a CO utilization of 23.7%. Importantly, the selectivity to linear esters, formed through an anti-Markovnikov addition, appeared to be independent of reaction time, temperature or pressure, accounting for approximately 75% of all carbonylation products.

Kinetic experiments were conducted to establish apparent activation energy and the reaction orders of CO and styrene, as illustrated in Supplementary Figs. 1–3. The reaction order for styrene was determined as 0.88, with the corresponding rate constant k of $2.9 \times 10^{-5}$ at 170 °C. Within the investigated CO pressure range (0.1–1 bar), the reaction order of CO is 0.76, with the rate constant k of $5.6 \times 10^{-10}$ (Supplementary Fig. 1). In a typical methoxycarbonylation reaction, the reaction was performed at a stirring rate of 600 rpm. Notably, the reaction rate was obviously low at 400 rpm, whereas the reaction rate at 600 rpm closely resembled that at 800 rpm, suggesting the elimination of potential mass transfer effects (Supplementary Fig. 2). The apparent activation energy of methoxycarbonylation reactions was determined as $60 \pm 11$ kJ/mol (Supplementary Fig. 3), basically aligning with the theoretical results (0.96 eV)[35].

Next, the effect of the size-dependent Ru speciation was investigated. The single Ru atom catalyst (Ru$_1$/NbO$_x$) and 4 wt.% Ru/NbO$_x$ catalyst with a diameter of 3.1 nm (Ru/NbO$_x$-3.1 nm) were synthesized (Supplementary Fig. 4 and Supplementary Table 11). Evaluated by the methoxycarbonylation of styrene, the activity of carbonylation presented a volcano curve with the size of Ru clusters (or nanoparticles). The 2 wt.% Ru/NbO$_x$ catalyst, featuring a diameter of 2.2 nm by CO-

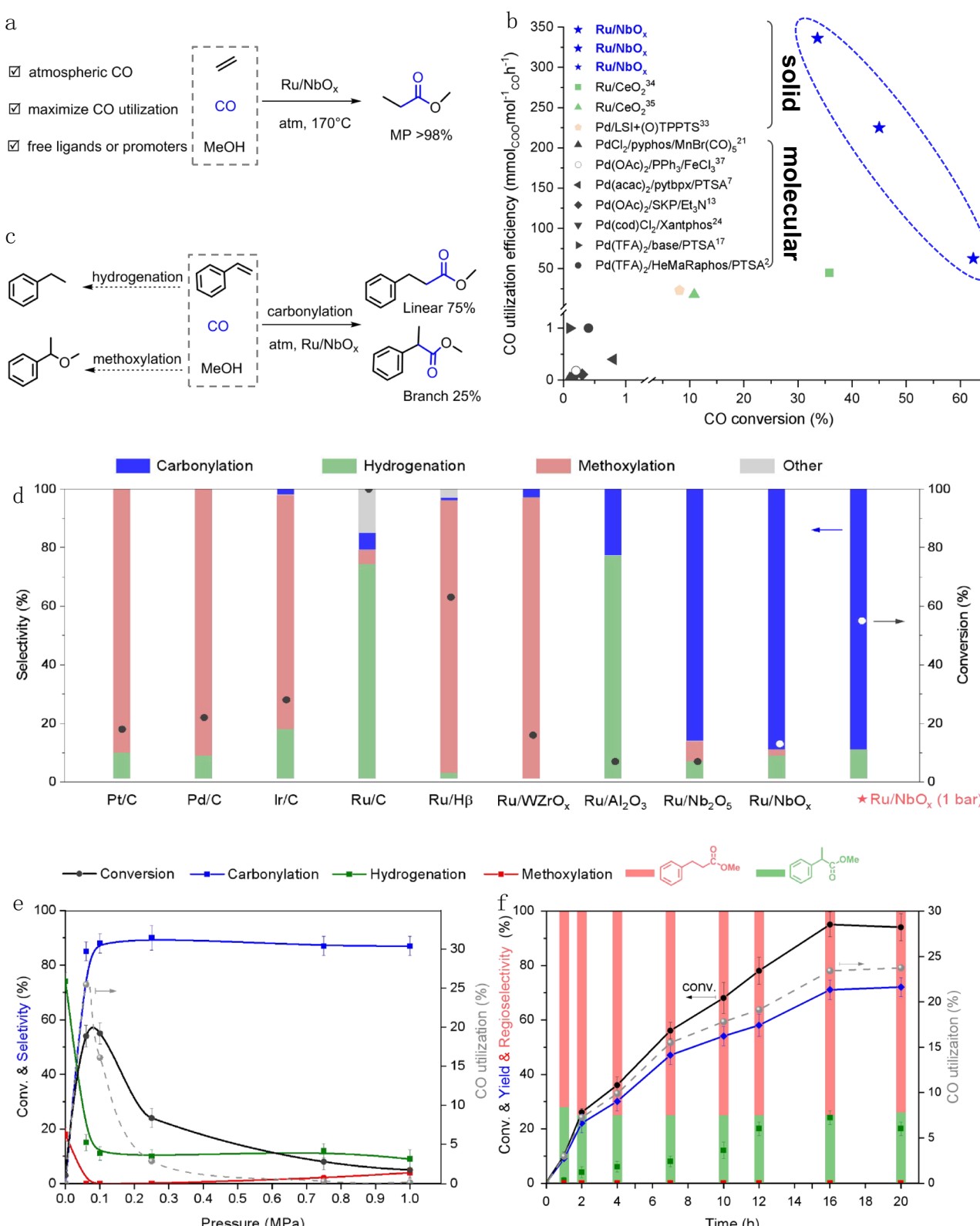

**Fig. 1 | Catalytic methoxycarbonylation of alkenes. a** Sustainable production of MP from ethylene and methanol under atmospheric pressure CO, with >98% selectivity of MP. **b** State-of-the-art cases reported in alkoxycarbonylation reactions of alkenes. Black symbols: molecular catalysts; colored symbols: solid catalysts. References are cited in (**b**), and for more details, see Supplementary Table 1. **c** Three possible pathways for the catalytic reaction involving styrene, CO and methanol.

The regioselectivity of carbonylation is shown. **d** Screening solid catalysts. Reaction conditions: 5 wt.% M/C (20 mg) or 2 wt.% Ru/oxide (30 mg), styrene (0.2 mmol), MeOH (2 mL), 160 °C, 10 h, CO (1 MPa) or ★ (0.1 MPa). Other is primarily the dimer of styrene. The reaction evolutions with **e** CO pressure and **f** time over Ru/NbOx. Reaction conditions: 2 wt.% Ru/NbOx (30 mg), styrene (0.2 mmol), MeOH (2 mL), 170 °C, 10 h in (**e**), 0.1 MPa CO in (**f**).

chemisorption, exhibited a better activity and carbonylation yield (62%), compared to the others tested (Supplementary Table 12). The $Ru_1/NbO_x$ showed that only <1% yield of hydrogenation was observed and no ester products were detected, most likely due to the absence of the accommodating reactive sites for multiple reactants. The $Ru/NbO_x$-3.1 nm exhibited a low carbonylation yield of 14% but a higher hydrogenation yield of 28%. The larger-size Ru nanoparticles favored the side-reaction hydrogenation of alkene substrates, most likely due to the enhanced metallicity.

## Catalyst structural characterization

To uncover the origin of the 2 wt.% $Ru/NbO_x$ catalysts' enhanced performance, its physicochemical properties were probed. High-resolution transmission electron microscopy (HRTEM) confirmed that this material is comprised of nanosheets with layers of exposed $Nb_2O_5$ (100) facets (Fig. 2a–c, Supplementary Fig. 5). To gain a more direct visualization of the metal distribution of 2 wt.% $Ru/NbO_x$, the aberration-corrected high-angle annular dark field scanning transmission electron microscopy (AC-HAADF-STEM) was performed. The images clearly demonstrated that ultrafine Ru nanoclusters were uniformly dispersed over the support (Fig. 3d–h). Quantitative electron paramagnetic resonance (EPR) was employed to identify (and quantify) defect sites. Analysis of the spectra showed that $NbO_x$ materials exhibited significantly higher concentrations of oxygen vacancy ($O_v$) sites compared to the analog $Nb(OH)_5$ and $Nb_2O_5$ (Fig. 2e–f). Interestingly, the $Ru/NbO_x$ catalyst possessed the highest concentration of $O_v$ sites, reaching up to $1.74 \times 10^{15}$ spins/g (Supplementary Fig. 6b). Furthermore, X-ray photoelectron spectroscopy (XPS) analysis of the O1s region confirmed that the $Nb_2O_5$ material exhibited a higher proportion of lattice oxygen ($O_L$), while $NbO_x$ had a greater concentration of defective $O_v$ compared to the other two (Fig. 2d). Thus, the findings from XPS corroborate our EPR analysis. Notably, the $Ru/NbO_x$ material exhibited a two-fold higher yield of esters (12%) than that of an analogous $Ru/Nb_2O_5$ catalyst (6%), thus accentuating the significance of $O_v$ sites in the carbonylation reaction (Fig. 2d).

The electronic and coordinative structures of 2 wt.% $Ru/NbO_x$ catalysts were further characterized by the X-ray absorption spectra (XAS) technique[41,42]. Figure 3a displayed the X-ray absorption near-edge spectra (XANES) at the Ru K-edge of the Ru catalyst and references. The adsorption threshold $E_0$ for $Ru/NbO_x$ is higher than Ru foil but lower than $RuCl_3$, suggesting the most likely coexistence of metallic and ionic Ru speciation, consistent with the XPS characterization results. Further, the coordination environment of Ru atoms was determined by the extended X-ray absorption fine structure spectra (EXAFS). As shown in the Fourier-transformed $k^2$- weighted EXAFS spectra at the Ru K-edge (Fig. 3b), in contrast to the reference samples of Ru foil and $RuO_2$, the Ru catalyst showed three major peaks, which could be ascribed to Ru-O, Ru-Ru, and Ru-O-Nb/Ru coordination. To further resolve Ru-O, Ru-Ru, and Ru-O-Nb/Ru coordination, wavelet transform (WT) of Ru K-edge EXAFS oscillations was carried out owing to its more powerful resolutions in both $k$ and $r$ spaces (Fig. 3c)[41]. The three hills at (1.45 Å, 5.00 Å$^{-1}$), (2.38 Å, 8.50 Å$^{-1}$), and (2.80 Å, 9.50 Å$^{-1}$), associated with Ru-O, Ru-Ru, and Ru-O-Nb/Ru contribution, respectively, was clearly observed from the WT contour plots of $Ru/NbO_x$ catalyst. Consistently, the best-fitted EXAFS result revealed Ru-O at 1.94 Å with coordination number (CN) of 3.6, Ru-Ru at 2.67 Å with coordination number (CN) of 5.1, and Ru-O-Nb/Ru shell at 3.08 Å with coordination number (CN) of 1.3, respectively, in $Ru/NbO_x$ (Supplementary Table 3 and Supplementary Fig. 7). The interfacial Ru-O-Nb-$O_v$ sites consist of Ru sites and intimate O or $O_v$ sites on the support. These sites are proposed to be synergistically active for the adsorptive dissociation of $CH_3OH$ on $O_v$ sites (Fig. 4a), CO adsorption exclusively on Ru sites (Supplementary Figs. 8–11), and alkene adsorption on both Ru and acidic Nb-$O_v$ sites (Figs. 4, 5 and Supplementary Figs. 9, 11), therefore promoting the subsequent hydromethoxycarbonylation

reaction of three substrates. We will discuss the role of the active sites in detail later.

## Potential reaction routes

The field of heterogeneous methoxycarbonylation is in its earliest infancy, with vast important unknowns yet to be explored. Besides developing an efficient catalyst, understanding the reaction pathway and mechanism is of high significance. First, the adsorption behaviors of methanol on the surface of Ru metal, the $NbO_x$ support and an $Nb_2O_5$ material (as a control), were investigated using density functional theory (DFT) calculations. The results demonstrated that methanol preferentially adsorbs onto $O_v$ sites on the $NbO_x$ surface ($G_{ads} = −0.96$ eV), compared to $Nb_2O_5$ ($G_{ads} = −0.58$ eV) or Ru ($G_{ads} = −0.61$ eV). Following this, the adsorbed $CH_3OH$ on $O_v$ sites can undergo dissociation to *$OCH_3$ and *H, with a remarkably low energy barrier of 0.1 eV, which is significantly lower than the 0.49 eV barrier observed over $Nb_2O_5$ (Fig. 4a). Methanol-probe Fourier transform infrared (FTIR) experiments support these findings, showing distinct O-$CH_3$ vibrations associated with *$OCH_3$ at 1030 cm$^{-1}$ on $NbO_x$, but absent on $Nb_2O_5$ (Supplementary Figs. 8, 9). The *$OCH_3$ and *H species formed can migrate from the $NbO_x$ support to the Ru surface with energy barriers of 0.28 eV and 0.77 eV, respectively (Fig. 4b). Complementary FTIR experiments were used to support this and showed increased O–$CH_3$ vibration signals over the $Ru/NbO_x$ (and $Ru/Nb_2O_5$) material(s), compared to the Ru-free support (Supplementary Figs. 9–11). These results evidenced that the $NbO_x$ support is responsible for the dissociation of $CH_3OH$ into *$OCH_3$ and *H species, which likely (at least partially) transferred to the Ru surface (or interface). The acidic sites of the $Ru/NbO_x$ catalyst were identified by pyridine-probe infrared spectra (Supplementary Fig. 12). The acidic $NbO_x$ support can facilitate the adsorption of alkenes, as evidenced by in situ infrared spectra (Supplementary Fig. 9). Additionally, CO exclusively adsorbs on the Ru surface with stronger adsorption energy (−1.65 eV) compared to $NbO_x$ (−0.23 eV), as confirmed by FTIR analysis (Supplementary Figs. 9, 11). Furthermore, styrene adsorption onto Ru surfaces is significantly more favorable ($G_{ads} = −3.46$ eV) than on $NbO_x$ ($G_{ads} = −0.98$ eV) (Supplementary Fig. 13).

These findings imply that the reactions involving styrene, CO, *$OCH_3$ and *H species most likely occur on the Ru surface. We subsequently explored four potential routes among styrene, CO, *$OCH_3$ and *H. They include (1) styrene carbonylation (styrene + CO), (2) styrene methoxylation (styrene + *$OCH_3$), (3) styrene hydrogenation (styrene + *H), and (4) methoxycarbonyl pathway (CO + *$OCH_3$). Each of the elementary reactions that could yield esters (both linear and branched) were considered. As shown in Fig. 5, (1) styrene carbonylation and (2) styrene methoxylation face significant barriers of approximately 1.20–1.60 eV, making the initial binding of styrene with CO or *$OCH_3$ unfavorable. For route (3) styrene hydrogenation, the energy barriers required for a *H species to attack an internal or terminal carbon in the side chain of styrene are low (0.59 eV and 0.86 eV, respectively). This suggests that *H species are more likely to interact with styrene ($PhCHCH_2$), particularly by hydrogenating its internal carbon atom to form *$PhCH_2CH_2$. However, when CO or *$OCH_3$ species attempt to react with this hydrogenated intermediate, they encounter large energy barriers that exceed 1.80 eV.

In the case of the route (4) methoxycarbonyl pathway, the first step involves the coupling of CO with a *$OCH_3$, resulting in the formation of *COOMe intermediate with a comparatively lower barrier of 1.06 eV ($\Delta G = 0.32$ eV). The crucial *COOMe species can selectively attack the terminal carbon in the side chain of styrene to form linear $C_8H_8COOMe$, facilitated by a lower energy barrier of 1.04 eV (TS2), while the production of branched $C_7H_6(COOMe)CH_2$ requires a higher barrier of 1.42 eV (TS2'). The terminal carbon atom of styrene is positioned 2.118 Å away from *COOMe in TS2 (Fig. 4c). Finally, the linear $PhCHCH_2COOMe$ intermediate undergoes hydrogenation to complete the cycle (a barrier

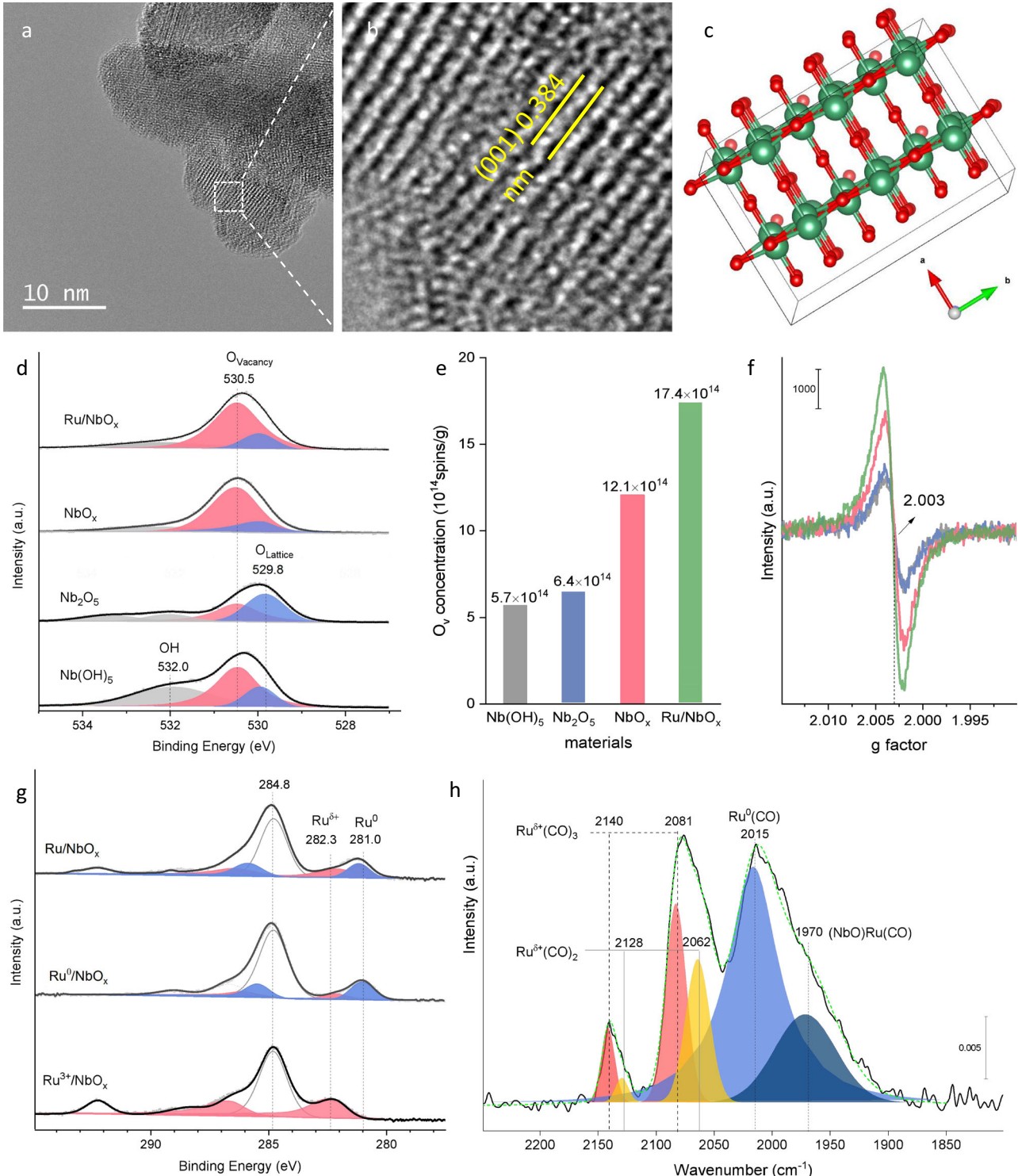

**Fig. 2 | Characterizations of Ru/NbOₓ and its analogs. a–c** HRTEM of 2 wt.% Ru/NbOₓ and crystal plane (001) model of Nb₂O₅. **d** O1s XPS. **e**, **f** Oxygen vacancy concentrations and EPR spectra. **g** Quasi-in situ Ru 3d XPS. **h** CO-probe FTIR of 2 wt.% Ru/NbOₓ.

of only 0.41 eV). Additionally, the formed *COOMe species can react with the hydrogenated *PhCH₂CH₂ or *PhCHCH₃ intermediates from styrene hydrogenation, but this step encounters large energy barriers of 1.99 eV and 1.32 eV, respectively (Fig. 5). Based on these calculations, we propose that the reaction proceeds via the methoxycarbonyl pathway, as shown in Fig. 4c. This involves the reaction of *CO and *OCH₃, leading to the formation of a *COOMe surface species, which can attack the terminal carbon in the side chain of styrene, leading to the formation of linear ester. From the perspectives of adsorption and reaction,

when *COOMe attacks the terminal C atoms of styrene (yielding linear esters), it experiences less steric hindrance, as well as a significantly lower energy barrier (1.04 eV). Besides, the secondary *C intermediate formed via the terminal attack is more stable than the primary *C species formed via the internal attack. In contrast, attacking the internal C atoms would encounter a much higher barrier (1.42 eV) to yield branched esters. These factors provide a rational explanation for the predominant linear regioselectivity observed in the methoxycarbonylation reaction over the Ru/NbOₓ catalyst.

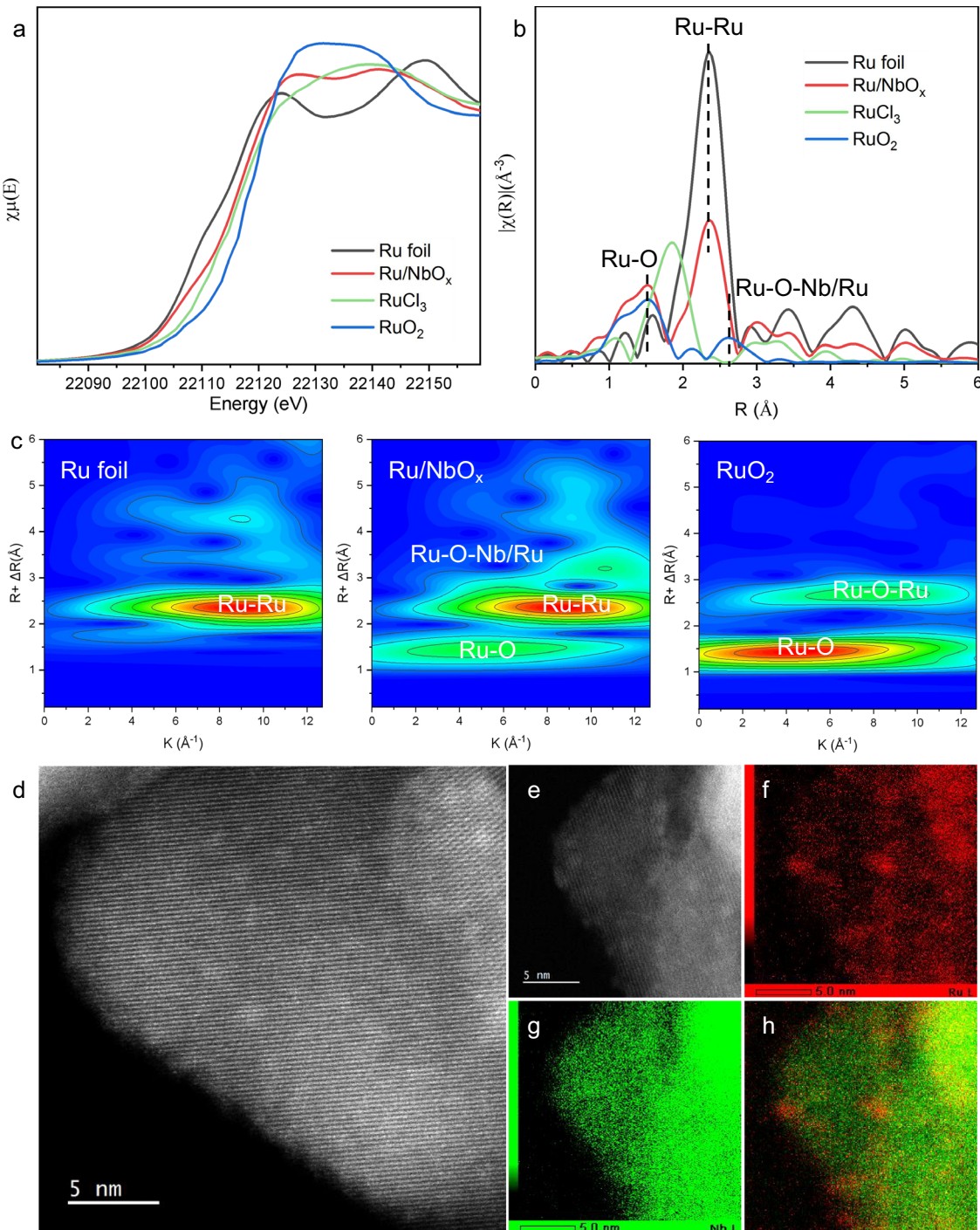

**Fig. 3 | Structural characterizations of 2 wt.% Ru/NbOₓ catalyst. a** The normalized X-ray absorption near-edge spectra (XANES) at the Ru K-edge. **b** The $k^2$-weighted Fourier transform extended X-ray absorption fine structure spectra (EXAFS) in r-space. **c** Wavelet transform of Ru foil, Ru/NbOₓ, and RuO₂. **d, e** AC-HAADF-STEM images and **f**–**h** the corresponding element map.

## Identification of heterogeneous methoxycarbonyl mechanism

In situ Fourier-transform infrared (FTIR) surface reactions were subsequently conducted to experimentally evidence the methoxycarbonyl pathway. The illustration scheme of the in situ infrared spectroscopic system applied in this work was depicted in Supplementary Fig. 15. Temperature-resolved FTIR experiments were conducted by saturating the catalyst with CO at 30 °C, followed by evacuating the cell. Subsequently, a methanol/styrene saturated vapor was introduced into the cell under vacuum until the system remained steady, and the infrared spectra were recorded. Stretching vibrational bands that are indicative of O–H, C=O, and $C_{Ar}$–$C_{Ar}$ bonds in methanol, CO, and styrene are observed at 3700, 2080, and 1450 $cm^{-1}$, respectively (Fig. 6a). As the temperature jumps from 30 to 60 °C, a new band attributed to C=O stretching vibrations emerged at 1750 $cm^{-1}$, exhibiting an increasing intensity over the temperature, indicating the gradual formation of *COOMe species[43,44]. The stretching vibration frequency of the C=O bond of *COOMe is further confirmed by the simulation using the Vienna Ab initio Simulation Package, at 1720 $cm^{-1}$ (Supplementary Fig. 16). The O–H vibrations of methanol at 3700 $cm^{-1}$ decreased by nearly 50% at 150 °C, due to both methanol consumption

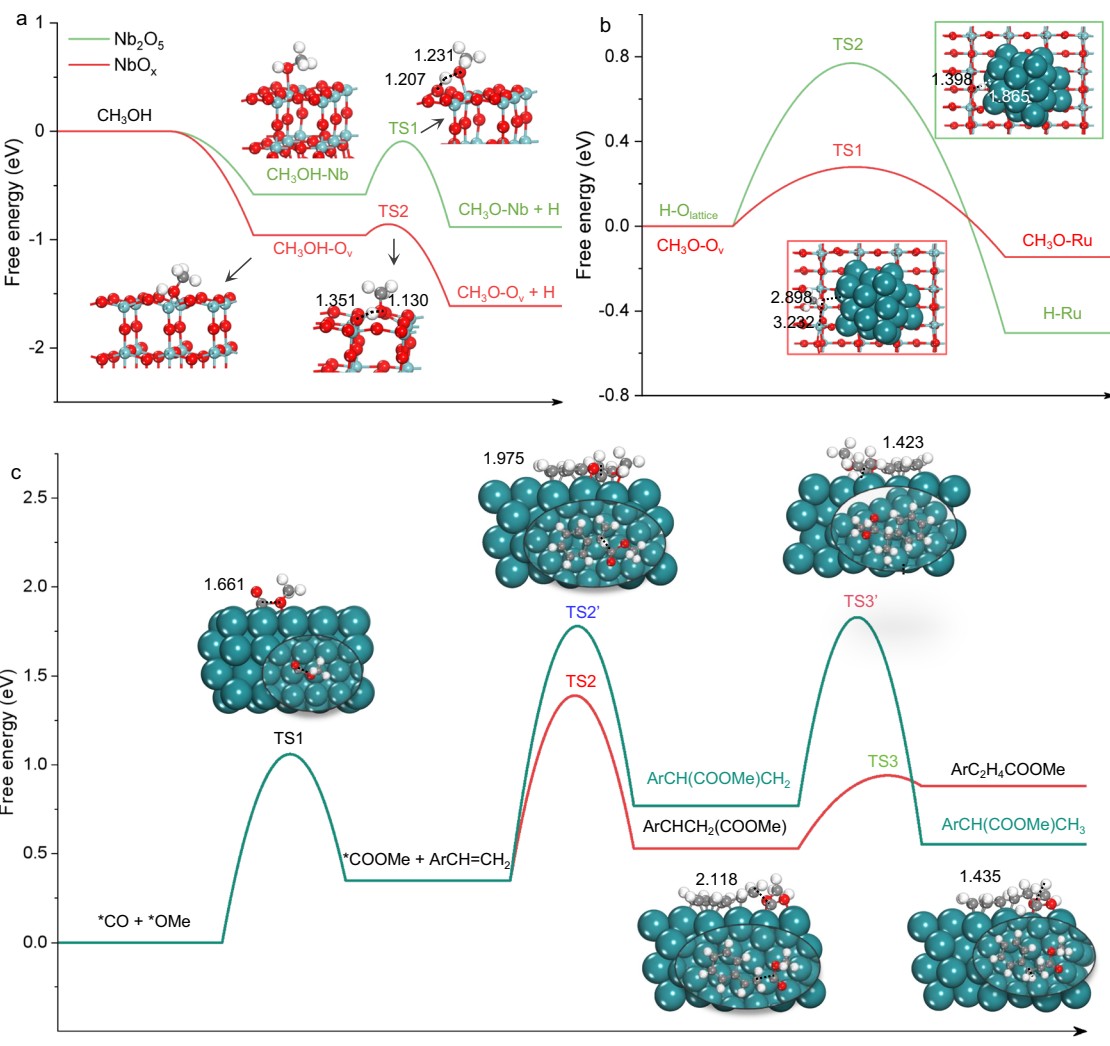

**Fig. 4 | DFT calculations on methanol dissociation and migration, and the methoxycarbonyl (COOMe) pathway to yield esters. a** Energy profiles for adsorptive dissociation of methanol on $Nb_2O_5$ and $NbO_x$, respectively. **b** Energy profiles for migration of *$OCH_3$ and *H species from $NbO_x$ support to Ru surface, respectively. **c** Energy profiles for the methoxycarbonyl pathway to yield linear- or branched-selective esters. The *x*-axis shows the reaction intermediates and transition states (TS); the *y*-axis shows the relative energy of each state. Nb, O, C and H atoms are shown in green, red, gray and white, respectively.

(for *COOMe formation) and heat-induced desorption. In contrast to the Ru/$Nb_2O_5$ analog, the Ru/$NbO_x$ catalyst showed a distinct peak at 1750 cm$^{-1}$, confirming its efficiency in generating *COOMe (Supplementary Figs. 10, 11). The time-resolved FTIR experiments were conducted at 150 °C to further investigate the surface species. The procedures were similar to the above experiment except that the experimental temperature was fixed at 150 °C. Following CO adsorption and evacuation in a vacuum, a methanol/styrene saturated vapor was introduced into the cell, and the infrared spectra were recorded. Immediately the vibrations at 1750 cm$^{-1}$ appeared and increased (by orders of magnitude) with time (Fig. 6b).

The Ru 3*d* XPS analysis of the Ru/$NbO_x$ catalyst revealed that the majority of the Ru species present on the material are in a metallic state (Ru$^0$), with only a minor proportion existing as Ru$^{n+}$ (Fig. 2g). In CO-probed FTIR spectra, the corresponding assignments of IR bands of Ru carbonyl species were assigned (Fig. 2h), based on the well-documented literature (Supplementary Table 14)[45–49]. Tricarbonyl or dicarbonyl Ru$^{δ+}$(CO)$_n$ species typically presented two high frequencies at ~2130 ± 15 and ~2070 ± 15 cm$^{-1}$, the monocarbonyl Ru$^0$(CO) species shows a frequency at 2030 ± 30 cm$^{-1}$. Two in situ FTIR experiments were conducted to gain further insights into the active Ru sites of the

catalyst. As shown in Fig. 6c, Ru/$NbO_x$ was saturated with CO at 150 °C, followed by the introduction of styrene vapor through argon bubbling. No vibration at 1750 cm$^{-1}$ corresponding to *COOMe was detected. Notably, the signals assigned to Ru$^{n+}$(CO)$_n$ at 2138 cm$^{-1}$ and 2072 cm$^{-1}$ decreased as styrene molecules were introduced. This decrease can be attributed to the partial occupation of the Ru surface by styrene. In the second experiment, Ru/$NbO_x$ was first saturated with CO before methanol vapor was introduced through bubbling with argon. The peaks at 1750 cm$^{-1}$ appeared and then continued to intensify over time, indicating that *COOMe was being formed through the combination of MeO* and *CO (Fig. 6d). Based on these findings, we consider that metallic Ru$^0$ and ionic Ru$^{n+}$ sites synergistically contribute to the catalytic process under hydrogen atmosphere (Supplementary Fig. 17). This was strongly supported by a significant fall in the carbonylation yield over the $NbO_x$ supported sole Ru$^0$ catalysts (Supplementary Table 15 and Fig. 2g). These infrared experiments strongly support the methoxycarbonyl pathway, and the results are consistent with the DFT calculations. According to this mechanism, methanol initially dissociates into *OMe and *H species on $NbO_x$, which then migrate to the Ru surface. CO adsorbed on the Ru surface first binds with *OMe to form the *COOMe intermediate, which subsequently attacks the

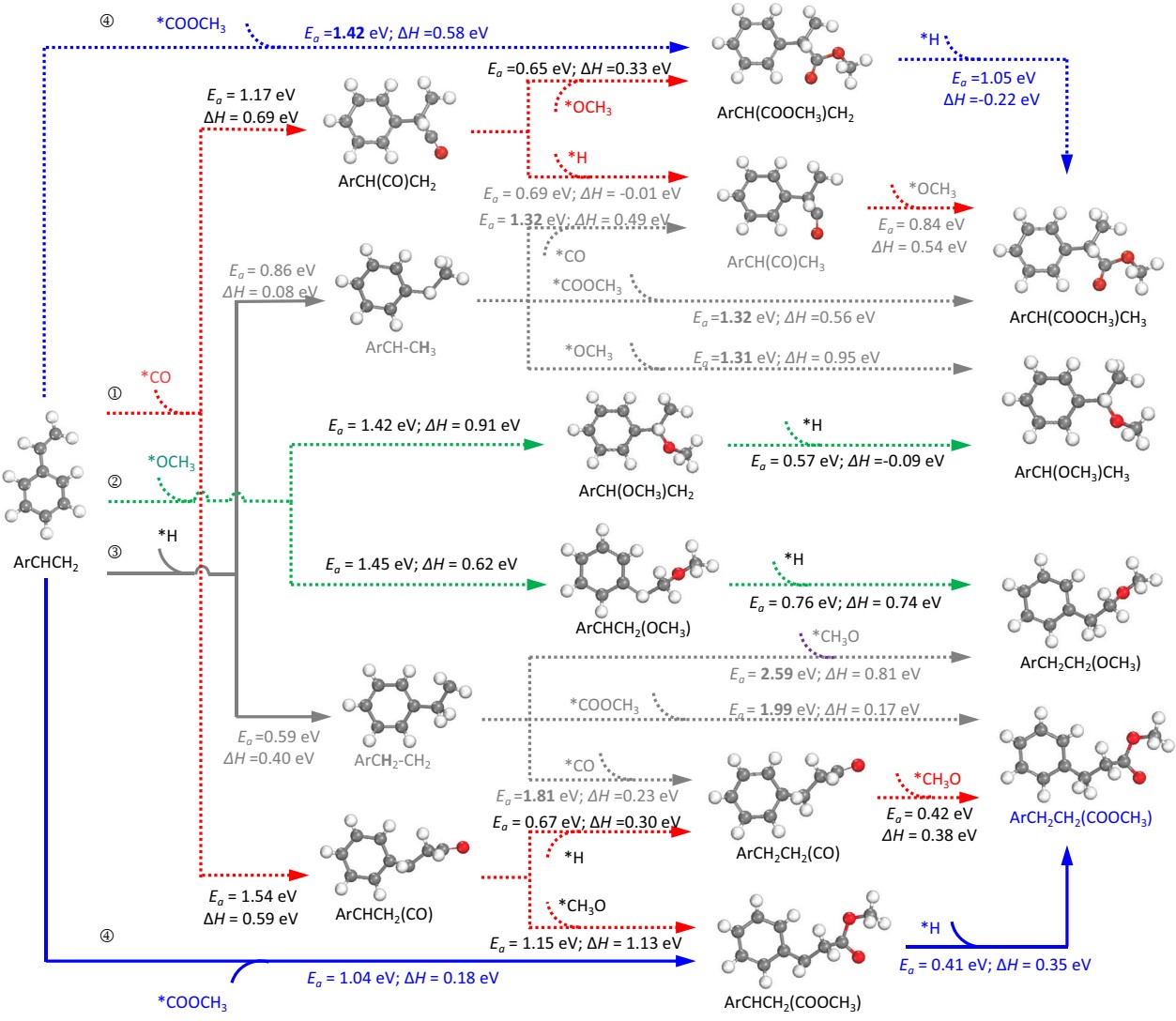

**Fig. 5 | The holistic elementary reaction networks of potential routes for the hydromethoxycarbonylation reaction are based on DFT calculations.** Four potential reaction routes to produce esters: ① styrene carbonylation (styrene + CO), ② styrene methoxylation (styrene + *OCH₃), ③ styrene hydrogenation (styrene + *H), and ④ methoxycarbonyl pathway (CO + *OCH₃). The energy barriers and enthalpy changes of each elementary step have been calculated and marked. The structures of the key intermediates are shown in Supplementary Fig. 14.

terminal carbon atom of the styrene side chain, followed by hydrogenation and desorption, as illustrated in Fig. 7.

### The substrate scopes and regioselectivity control

To assess the generality of the catalysts, a series of other substrates were also investigated under the optimized reaction conditions (Supplementary Tables 16–20). The Ru/NbO$_x$ catalyst also exhibited activity for styrene derivatives with *para*-position substituents such as acetoxyl, bromo, methoxy, hydroxyl, and tertiary butyl, resulting in an ester yield of ca. 50% (Supplementary Table 16). All three α-olefins with vinyl, propenyl, and butenyl side chains achieved conversion of >96%, while styrene showed the highest carbonylation product yield of 71%, due to its inherent lower steric hindrance (Supplementary Table 17). To explore the influence of C=C positions in the substrate on carbonylation, the reactivity of allyl and propenyl benzene was assessed. Interestingly, similar product distributions were obtained regardless of the C=C positions, primarily due to preferential isomerization (Supplementary Tables 18, 19). Among the four alcohols examined (methanol, ethanol, propanol, and butanol), methanol demonstrated the highest selectivity to alkoxycarbonylation products (Supplementary Table 20). Controlling regioselectivity (toward Markovnikov or anti-

Markovnikov) is evidently crucial for the efficient carbonylation of alkenes. We speculate that the acid/alkalinity of the environment affects the stabilization of primary/secondary intermediate and, thus, influences the regioselectivity. To probe this, some additional experiments were conducted where trace quantities of base or acid were spiked into the styrene methoxycarbonylation reaction. The results show that the presence of basic CsCO₃ or Me₃EtOK species, results in exclusive regioselectivity to branched esters via Markovnikov addition. On the contrary, anti-Markovnikov additions resulting in the formation of linear esters accounted for 75% of the products in both acidic and neutral environments (Supplementary Fig. 18).

Despite the advantages of heterogeneous Ru/NbO$_x$ catalyst, there remains an issue of cycling stability (Supplementary Table 21). Comprehensive characterization of the fresh and cycled catalysts from the same batch was conducted (Supplementary Fig. 19). Pyridine-probe FTIR revealed a significant decline (ca. 50% or more) in Brönsted acid sites for the used catalyst compared to the fresh one, while the amount of Lewis acid sites remained nearly unchanged. CO-probe FTIR indicated a substantial decrease in exposed Ru species for the used catalyst, suggesting a potential structural change of Ru surface during the reaction (Supplementary Fig. 19b). The AC-HAADF-STEM images of the

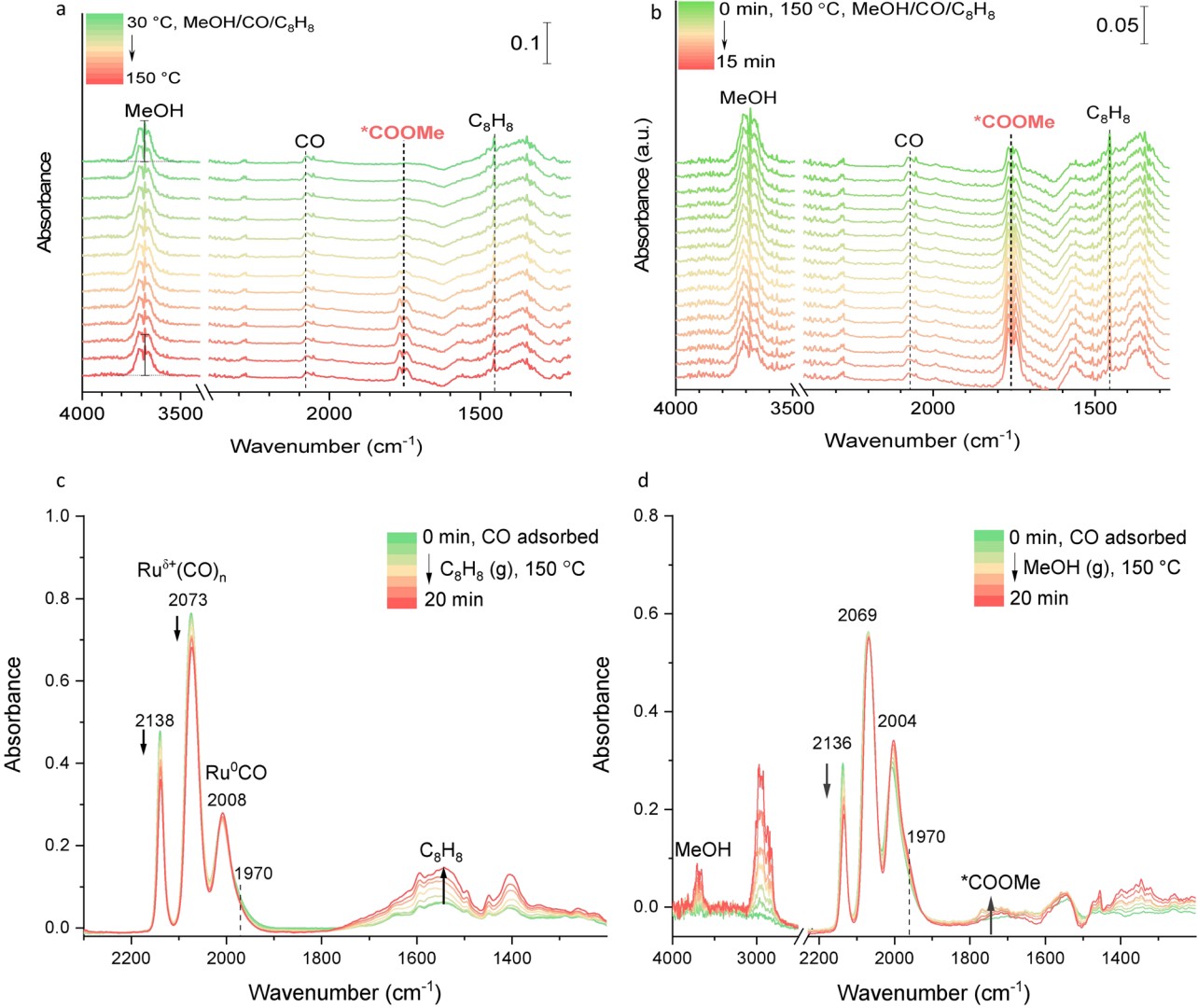

**Fig. 6 | Experimental evidence for the methoxycarbonyl species formation.** In situ FTIR surface reactions of CO, styrene and methanol over Ru/NbOₓ: **a** temperature-resolved reaction after chemisorption of CO and methanol/styrene vapor. **b** time-resolved process at 150 °C. **c** adsorption of styrene vapor after CO desorption at 150 °C. **d** adsorption of methanol vapor after CO desorption at 150 °C.

used catalyst exhibited prevalent particulate Ru aggregates, while no highly dispersed Ru clusters were observed in the investigated regions (Supplementary Fig. 20). ICP-OES results also showed the loading of Ru metal slightly declined from 1.89% to 1.28%. Furthermore, EPR analysis of fresh and used catalysts was carefully measured and confirmed that the number of oxygen vacancies in the used catalysts did not decline after the reaction (Supplementary Fig. 19c). Ru 3d XPS results showed that Ru/NbOₓ catalyst, both before and after the reaction, contained both Ru$^{\delta+}$ and Ru$^0$ components with a slightly changed proportion (Supplementary Fig. 19d). Based on these analyses, we can conclude that the agglomeration of highly dispersed Ru clusters over support is most likely responsible for the observed deactivation. Given the significance of this issue, we aim to address it to improve recyclability in the near future.

## Results and discussion

We have developed a highly effective and sustainable catalytic system for the methoxycarbonylation of alkenes under ambient pressure. The methoxycarbonyl mechanism was revealed, that is, methanol initially dissociates into *OMe and *H species on Oᵥ sites of NbOₓ, which then migrate to the Ru surface. CO adsorbed on the Ru surface first binds with *OMe to form the *COOMe intermediate, which subsequently

attacks the terminal carbon atom of the styrene side chain, followed by hydrogenation and desorption, yielding linearly regioselective esters. Importantly, our investigations determined that, under basic conditions, the reaction exclusively produces branched esters (via a Markovnikov reaction). On the contrary, under acidic and neutral environments, linear esters were predominantly produced (via an anti-Markovnikov reaction). Notably, the sustainable production of MP was obtained in a yield of 60% (with 98% selectivity) on the Ru/NbOₓ catalyst. Overall, this discovery opens new possibilities for fine-tuning regioselectivity in alkoxycarbonylation reactions, broadening the application of heterogeneous alkoxycarbonylation, and providing valuable insights into the development of sustainable catalytic systems for ester production.

## Methods

### Chemicals

Niobium (V) oxalate hydrate (99.0%, Alfa Aesar), cetyltrimethylammonium bromide (CTAB, 99.0%, Sigma-Aldrich), RuCl₃ (>99.0%, TCI), styrene (>99.5%, 3AChem), methanol (>99.9%, Innochem), n-dodecane (>99.0%, Alfa Aesar), methyl propionate (99%, J&K), 3-phenylpropionic methyl ester (>98.0%, Macklin), 2-phenylpropionic methyl ester (>98.0%, Macklin), (1-methylpropane-

a **Hydromethoxycarbonylation for sustainable ester production**

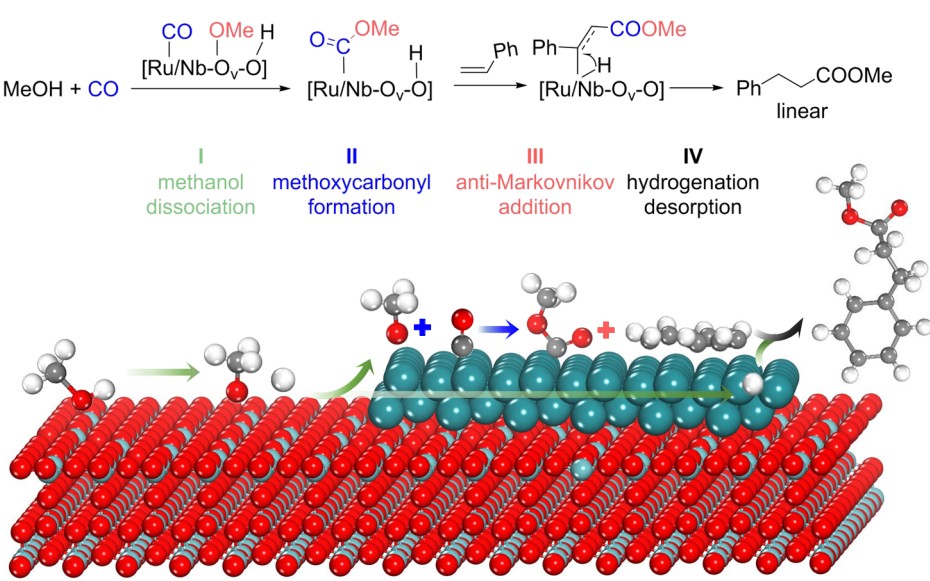

b **Heterogeneous methoxycarbonyl (COOMe) pathway**

**Fig. 7 | Heterogeneous alkoxycarbonylation reactions. a** The reaction route. **b** The methoxycarbonyl mechanism.

1,3-diyl)dibenzene (>98.0%, Macklin), 4-hydroxystyrene (>95.0%, J&K), 4-methoxystyrene (>98.0%, 3AChem), allylbenzene (>98.0%, Innochem), β-methylstyrene (98.0%, Alfa Aesar), 4-allylanisole (>97.0%, Aladdin), trans-anethole (>98.0%, Innochem), (2-methoxyethyl)benzene (>98.0%, Macklin), 4-phenyl-1-butene (98.0%, Aladdin), sodium tetraborate (>99.0%, 3AChem), ammonium fluoride (>99.8%, Macklin), cesium carbonate (99.9%, Alfa Aesar), potassium benzoate (99.0%, Alfa Aesar), potassium acetate (99.0%, Acros). 10%H$_2$/Ar (>99.999%) and Ar (>99.999%) were supplied by Beijing Analytical Instrument Company. All chemicals were used without further purification.

### Synthesis of NbO$_x$ and Nb$_2$O$_5$ support
For this, 5.38 g of niobium oxalate hydrate and 1 g of CTAB were dissolved in 30 mL of water and then transferred to hydrothermal treatment at 180 °C for 10 h. After cooling, the white precipitate was washed with water until no foam was observed, and then washed with ethanol and dried in an oven at 60 °C for 12 h. The obtained powder was reduced at a 10% H$_2$/Ar atmosphere at 550 °C for 4 h, denoted as NbO$_x$. If the calcination atmosphere was air, the resulting sample was labeled as Nb$_2$O$_5$.

### Synthesis of 2 wt.% Ru/NbO$_x$, Ru/Nb$_2$O$_5$, Ru$^0$/NbO$_x$, Ru$^{3+}$/Nb$_2$O$_5$, and Ru$_1$/NbO$_x$ catalysts
For this, 510 μL of RuCl$_3$ aqueous solution (0.077 mol/L) was dropwise added to 200 mg NbO$_x$ powder in 2 mL water and stirred for 24 h at room temperature. The solution was dried at 120 °C for 2 h, followed by calcination at 200 °C for 2 h with a heating rate of 5 °C/min in a muffle oven. The obtained powder was ground and reduced in 10% H$_2$/Ar at 350 °C for 4 h with a heating rate of 5 °C/min in a tubular oven, denoted as Ru/NbO$_x$. Ru/Nb$_2$O$_5$ was synthesized using the same procedure except for employing Nb$_2$O$_5$ as support. Ru$^0$/Nb$_2$O$_5$ was prepared using the same process except for a reduction

temperature of 550 °C. Ru$^{3+}$/Nb$_2$O$_5$ was prepared using the same procedure but no calcination and reduction procedures. Ru$_1$/NbO$_x$ was synthesized using the same procedure except for 0.1 wt% Ru loading.

### Characterization
Powder X-ray diffraction analysis was performed at a Smart Lab X-ray diffractometer using Cu-Kα radiation (λ = 0.15432 nm), operating at 40 kV and 40 mA. High-resolution transmission electron microscopy images were obtained using a JEOL-2100F electron microscope operated at 120 kV. Electron paramagnetic resonance (EPR) was characterized by Bruker A300-10/12. X-ray photoelectron spectroscopy (XPS) was conducted on an X-ray photoelectron spectrometer (USA, Thermo Fischer, ESCALAB 250Xi) equipped with a monochromatized Al Kα excitation source (1486.8 eV), using C1s (284.8 eV) of adventitious carbon as the standard. Quasi-in situ XPS experiments were performed by loading the freshly reduced sample to the XPS sample holder in a glove box and then transfering into an ultra-high-vacuum chamber for XPS measurement. The actual Ru loadings of Ru-based catalysts were determined by inductively coupled plasma optical emission spectrometer (ICP-OES) through microwave-assisted digestion treatment. CO-pulse chemisorption experiments of Ru/NbO$_x$ catalysts were conducted on AutoChem II 2920. Before performing the measurement, the catalyst sample was reduced in 10% H$_2$/Ar at 350 °C for 1 h with a heating rate of 10 °C/min and purged with flowing helium for 0.5 h until the baseline was stabilized. Ru dispersion (D%) of Ru-based catalysts was evaluated by CO-pulse chemisorption and calculated by Eq. (1)[50].

$$D\% = \left( \frac{V_{co} \times AF_{Ru/CO} \times M_{Ru}}{22414 \times m_{cat} \times wt.\%_{Ru}} \right) \times 100\% \qquad (1)$$

The $V_{CO}$ is the accumulative adsorption volume of CO (mL), $M_{Ru}$ is the atomic weight of ruthenium (101.1 g/mol), $m_{cat}$ is the mass weight of the catalysts (g), wt.%$_{Ru}$ is the actual Ru weight percentage of the catalysts. $AF_{Ru/CO}$ is the average factor of $n_{Ru}/n_{CO}$ (Supplementary Table 13).

The in situ Fourier transform infrared surface reaction (FTIR) experiments were performed in a custom-designed FOLI10-R-T infrared spectrometer equipped with dual sample chambers (transmission cell and diffusion reflection cell) and dual detectors (DLaTGS detector made by INSA Co. Ltd. and operando snail cell made by Beijing Operando Technology Co., Ltd). The FTIR was operated with a transmission operando snail cell at a resolution of 4 cm$^{-1}$ and scanned at a speed of one spectra per minute. In a temperature-resolved FTIR experiment, 20 mg of catalyst was ground and pressed into a self-supported wafer with 13 mm diameter in a mold, followed by transfer to operando snail cell. It was then treated in 10% H$_2$/Ar at 350 °C for 30 min with a heating rate of 10 °C/min. Once cooling, the atmosphere was switched to Ar (40 mL/min). The spectra were continuously recorded at 30 °C until the signal was kept unchanged, and then acquisited one as the background spectra. The pure CO gas was introduced into the reaction cell until the spectra were unchanged. Subsequently, pure Ar (40 mL/min) was purged to remove the free CO until the spectra were unchanged. The methanol/styrene (1/10 V/V) mixture was bubbled by Ar (40 mL/min), and the spectra were recorded continuously until the signal stabilized. The temperature was programmed to the desired temperature with a heating rate of 10 °C/min, and collected the spectra per minute continuously. For the time-resolved FTIR experiments, the measurement employed the same process except for the constant temperature of 150 °C during the whole period of spectra acquisition after the hydrogen reduction pretreatment.

## Alkoxycarbonylation reaction evaluation

The reaction was performed in a stainless steel reactor with a 16 mL Teflon liner. In a typical experiment, the given amount of catalyst, n-dodecane, methanol and substrate were loaded into the reactor. The reactor was sealed and purged with pressured CO until the desired gauge pressure (as stated throughout this work; ambient pressure refers to 0 Pa of gauge pressure) was achieved. Then, the reactor was placed in a heating oven and stirred at 600 rpm. Once finished, the reactor was immediately cooled down in flowing tap water. Following that, the reactor was slowly released, and the resultant mixture was separated by centrifugation (13,000 rpm for 1 min). The quantitative analysis of the liquid products was conducted using a GC (Agilent 6820) equipped with a flame ionization detector and an HP-5 capillary column (0.25 mm in diameter, 30 m in length). Identification of the products and reactants was conducted by standard reagents in GC by comparing the retention time with n-dodecane as the internal standard in GC tests and performed in a GC–MS equipped with an HP-5 capillary column. The carbon balance was based on the total amount of benzene rings. To calculate the conversion of styrene and the selectivity of esters, an internal standard method based on the GC data was employed. The GC spectra for ethylene methoxycarbonylation are shown in Supplementary Fig. 21.

CO utilization (rate) is defined as the ratio of the amount of ester produced to the total amount of CO and is calculated by Eq. (2). p, V, R and T represent the actual CO pressure (Pa), volume (14 × 10$^{-6}$ m$^3$), molar gas constant (8.314 J·K$^{-1}$·mol$^{-1}$) and temperature (K). The p (Pa) equals the sum of gauge pressure (p$_g$, Pa) and atmospheric pressure (p$_0$, 10$^5$ Pa).

$$CO\ utilization = \frac{n_{ester}}{n_{CO}} \times 100\% = \frac{n_{ester}}{pV/RT} \times 100\% \quad (2)$$

CO utilization efficiency (h$^{-1}$) is defined as the CO utilization rate per unit time (h) and calculated by CO utilization/t (h).

## DFT calculations

All the spin-polarized density functional theory (DFT) calculations were performed with the VASP code[51], using the Perdew-Burke-Ernzerhof (PBE) functional within the generalized gradient approximation (GGA)[52]. The core-valence electron interaction was represented by the project-augmented wave (PAW) method[53]. The valence electrons were expanded in a plane-wave basis set with a cutoff energy of 450 eV. The Broyden method was employed for geometry optimization until the maximal force on each relaxed atom was less than 0.05 eV/Å. The transition states (TSs) were searched by the constrained minimization method[54]. The empirical correction in Grimme's scheme was used to describe the van der Waals interactions[55]. Both Ru(101) and Nb$_2$O$_5$(001) surfaces were modeled by a four-layer slab model with a vacuum of 15 Å, which correspond to a k-points mesh of 2 × 2 × 1 and 2 × 2 × 1, respectively. The choice of Nb$_2$O$_5$ (001) was justified by its presence as the most stable crystal facet in the XRD pattern (Supplementary Fig. 6) and its clear visibility in the HRTEM images (Fig. 2a, b). The *hcp* Ru (101) was selected for the crystal facet stability, known as the most stable facet for Ru, based on XRD diffraction patterns and literature references[38]. During structural optimization, the bottom two layers of the surface were fixed at the bulk truncated position, and the top two layers and the adsorbates were fully relaxed. The adsorption energy ($E_{ads}(X)$) of adsorbate X on the surface was calculated with the equation:

$$E_{ads}(X) = E_{x\_surf} - E_{surf} - E_x \quad (4)$$

where $E_x$, $E_{surf}$ and $E_{x\_surf}$ are the total energies of the adsorbate X in the gas phase, clean surface and the surface with adsorbate X, respectively. The negative $E_{ads}(X)$ indicates the stronger adsorption strength. Noteworthily, the Gibbs free energy change ($\Delta G$) and the barrier ($E_a$) of the elementary step were estimated ($\Delta G = \Delta H + \Delta E_{ZPE} - T\Delta S$), including the thermodynamic zero point energy (ZPE) correction and the entropy contribution ($T\Delta S$). Regarding $T\Delta S$ and $\Delta E_{ZPE}$, we calculated the vibrational frequencies of the surface intermediates and transition states within DFT calculation[56]; for the gaseous molecules, the entropy contributions ($T\Delta S$) were derived from the experimental values reported.

## Data availability

The structural data of the calculations, including the optimized xyz structural coordinates of the crucial intermediates and transition states, the corresponding total energy, thermodynamic zero point energy (ZPE) correction and the entropy contribution (T$\Delta$S) are shown in Supplementary Note 7. The data that support the findings of this work are available within the manuscript and Supplementary Information files, all of which are available from the authors on request.

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

## Acknowledgements

This work received financial support from the National Key Research and Development Program of China (2022YFA1504901, 2022YFA1504903) and the National Natural Science Foundation of China (22302209, 22293012, 22293015, 22003069, 22179132, 22121002), and Junior Fellowship of Beijing National Laboratory of Molecular Science (2021BMS20059), and Marie Sklodowska-Curie (UKRI) Postdoctoral Fellowships (EP/X021734/1). M.D., N.F.D., R.J.L. and G.J.H. gratefully acknowledge Cardiff University and the Max Planck Centre for Fundamental Heterogeneous Catalysis (FUNCAT) for financial support. B.Z. thanks Dr. Ke Tian from the Institutional Center for Shared Technologies and Facilities, Institute of Process Engineering, Chinese Academy of Sciences, for EPR characterizations and analysis. B.Z. thanks Dr. Haifeng Qi for his contributions in XAS characterizations and analysis.

## Author contributions

B.Z. and H.L. conceived the project and designed the experiment. B.Z. and Y.L. conducted the experiments, and H.Y. conducted and analyzed the theoretical calculations. Z.D., M.D., N.D., R.L., X.L., S.L., M.D., T.W., Q.X., Z.Z., B.X. and G.H. provide constructive ideas to the experiment results. B.Z., H.Y. and H.L. wrote the manuscript.

## Competing interests

The authors declare no competing interests.
