## [Peer Review File · Nature Communications]

Ambient-pressure Alkoxy-carbonylation for Sustainable Synthesis of EsterREVIEWER COMMENTS

Reviewer #1 (Remarks to the Author):

This manuscript outlines a novel solid catalyst exhibiting pronounced (regio)selectivity in the alkoxy carbonylation of olefins. The catalyst, comprising Ru nanoparticles on partially reduced Nb₂O₅, achieves elevated CO conversions and reactant utilization rates, even under lower CO partial pressures when compared to alternative solid and technical benchmark molecular catalysts. Notably, this catalyst demonstrates remarkable efficiency in converting ethylene, CO, and methanol to methyl propionate, showcasing its industrial relevance.

While ethylene alkoxy carbonylation is the industrially most relevant transformation, the mechanistic studies presented in this manuscript predominantly employ styrene (and its derivatives) as the olefin substrate. The outcomes underscore the pivotal role of the support nature and the Ru/support interface in dictating performance. These findings bear significant relevance in the catalysis domain. All in all, the results presented herein contribute significantly to the ongoing effort to transition from molecular catalysts in solution to solid catalysts—a noteworthy challenge in catalysis. The finds presented, if refined, could serve as a foundation for publication in Nature Commun. Nevertheless, certain aspects of the manuscript need revisiting to strengthen specific claims. Please refer to the remarks below.

1) The observation that lower CO partial pressures lead to higher CO conversions in batch-type tests is unsurprising. However, I question the relevance of the CO conversion level when the olefin is the limiting reactant. In my view, the more pertinent figure of merit for the system is CO utilization rather than conversion.

2) The commendable performance of the Ru/NbO_x catalyst under comparatively lower CO partial pressures suggests distinctive reaction kinetics compared to comparative catalysts. To shed light on these differences, I recommend the experimental determination and reporting of the reaction orders for CO and methanol for both the newly developed solid catalyst and a relevant benchmark catalyst.

3) Anticipating a lower driving force for CO solubilization in the liquid reaction medium under lower CO partial pressures, I suggest a thorough assessment of the kinetic significance of mass transfer steps under the applied reaction conditions. This is important to corroborate that the rates determined experimentally faithfully correspond to the hydroxy carbonylation reaction.

4) DFT methods: The description of the DFT methods lacks sufficient detail. Additional elaboration on the DFT methods employed in this study is necessary to provide a comprehensive description of the computational approach. The rationale behind selecting Ru(101) and Nb₂O₅(001) facets for constructing the slab models in DFT calculations needs clarification too.

5) Further elaboration is needed on the differences in the slab models developed for Ov-NbO_x (partially reduced) and Nb₂O₅. It is difficult to perceive differences between the models as well as the optimized structures for the reaction transition states in the figures. Hence, I suggest to add all structural data of the calculations, including optimized xyz coordinates and corresponding total energy, to the Supplementary Information.

6) Presenting the predicted free energy diagram at the relevant reaction temperature would be more insightful than the E diagram. The reaction mechanism presented as most feasible has a reaction barrier of 1.15 eV. This appears to me a rather high barrier for a reaction which proceeds with relevant TOF at temperatures around 150°C. Can the authors compare this to an experimentally obtained apparent activation energy?

7) In situ FTIR spectroscopy: the discussion requires significant clarifications. Authors indicate in the text "then introducing a methanol/styrene solution into the cell". However, the methods section suggests introduction of reactants in the vapor phase. This should be clarified. Please provide schemes of the cell system applied for these spectroscopic studies in the ESI.

8) The band assignment remains unclear and questionable, particularly with single IR bands being assigned to polycarbonyl ($\text{Ru}+(\text{CO})_n$) species. Addressing the discrepancy concerning the expected multiple C-O stretching vibrations (symmetric and asymmetric) for polycarbonyls is essential.

9) The ascription of the C-O stretching vibration to Ru atoms interfaced with niobia requires further explanation. Why would the authors expect a higher degree of d- π^* metal-to-CO backdonation for interfacial sites? I encourage the use of DFT methods to predict vibrational frequencies for relevant adsorbates and various sites to support band assignments.

10) The spectroscopic features associated with *COOMe species, crucial for supporting the mechanistic proposal, appear remarkably weak, fundamentally undiscernible from the spectral background. The adequacy of these features in conclusively proving the development of these reaction intermediates should be addressed.

11) While the assessment of Ru dispersion on different catalysts based on CO chemisorption is noted, providing a rationale for the chemisorption stoichiometry factors considered in dispersion calculations is required. Additionally, X-ray absorption spectroscopy is suggested for an independent and more reliable assessment of the average metal dispersion.

12) Notation: referring to "solid" and "molecular" catalysts rather than "heterogeneous" and "homogeneous" catalysts, respectively, is proposed. The latter adjectives are more appropriately applied to "catalysis" than to the "catalysts" involved.

Reviewer #2 (Remarks to the Author):

The manuscript "Ambient-pressure Alkoxy carbonylation for Sustainable Synthesis of Ester" demonstrates the use of Ru supported over NbO_x for alkoxy carbonylation of olefins at ambient pressure of CO. This work is interesting in terms of studying the mechanism of the reaction using in-situ FTIR and DFT. However, I am not convinced regarding its higher efficiency compared to earlier published heterogeneous catalysts. Firstly, the term CO utilization, to which the authors refer, does not seem very important for comparison. It increases at low pressure but also depends on the volume of the reactor. Thus, it is possible to significantly increase it by using a small volume reactor. The earlier published Ru/ CeO_2 catalyst (Chinese Journal of Catalysis 41 (2020) 963–969, J. Am. Chem. Soc. 2018, 140, 11, 4172–4181) also demonstrates high activity and selectivity under similar conditions and shows an increase in activity at atmospheric pressure. Therefore, I would recommend comparing the performance with this catalyst under the same conditions in the same reactor. This would provide a clearer vision about the advantage of NbO_x as a support. Although there are some mechanistic studies, I wouldn't say that this work is sufficiently new to be published in Nat Com. I think it would be better suited to a more specialized journal. Additionally:

1. I recommend performing a comparison with already published O-deficient supports, such as CeO_2 , to demonstrate the efficiency of Ru/ NbO_x . I also couldn't find activity over Ru/ Nb_2O_5 and reference samples in Figure 2.

2. There are no results regarding the stability of the catalyst for several cycles. Most probably, O

vacancies are deactivated in the presence of methanol, leading to a decrease in activity over time.

3. The mechanism proposed by the authors involves the conversion of $\text{Ru}^+(\text{CO})_n$ species to metallic Ru with the formation of $^*\text{COOMe}$ intermediate, indicating the synergetic role of Ru^+ and Ru metallic. However, this would mean that the reaction is not catalytic, or the mechanism of regeneration of cationic Ru should be proposed. It seems unclear.

4. Nb oxide is the most acidic oxide support. What is the role of these acid sites in the reaction? Is it possible that these sites could stabilize Ru cationic? Or they are involved in adsorption of ethylene according to the mechanism proposed earlier for this reaction.

5. Earlier it has been found that Ru/CeO₂ contains Ru-O-Ce sites (J. Am. Chem. Soc. 2018, 140, 11, 4172–4181), what's about Ru-O-Nb in this catalyst?

6. It would be important to provide more information about the structure sensitivity of the catalyst. What should be the size of Ru clusters for optimal performance? How does activity change depending on the size of the clusters?

REVIEWER COMMENTS

Reviewer #1 (Remarks to the Author):

This manuscript outlines a novel solid catalyst exhibiting pronounced (regio)selectivity in the alkoxy carbonylation of olefins. The catalyst, comprising Ru nanoparticles on partially reduced Nb₂O₅, achieves elevated CO conversions and reactant utilization rates, even under lower CO partial pressures when compared to alternative solid and technical benchmark molecular catalysts. Notably, this catalyst demonstrates remarkable efficiency in converting ethylene, CO, and methanol to methyl propionate, showcasing its industrial relevance. While ethylene alkoxy carbonylation is the industrially most relevant transformation, the mechanistic studies presented in this manuscript predominantly employ styrene (and its derivatives) as the olefin substrate. The outcomes underscore the pivotal role of the support nature and the Ru/support interface in dictating performance. These findings bear significant relevance in the catalysis domain. **All in all, the results presented herein contribute significantly to the ongoing effort to transition from molecular catalysts in solution to solid catalysts—a noteworthy challenge in catalysis. The finds presented, if refined, could serve as a foundation for publication in Nature Commun.** Nevertheless, certain aspects of the manuscript need revisiting to strengthen specific claims. Please refer to the remarks below.

Response: We appreciate your strongly supportive recognition of our work, particularly in tackling the challenge of alkoxy carbonylation using solid catalysts.

1) The observation that lower CO partial pressures lead to higher CO conversions in batch-type tests is unsurprising. However, I question the relevance of the CO conversion level when the olefin is the limiting reactant. In my view, the more pertinent figure of merit for the system is CO utilization rather than conversion.

Response: We thank you for your valuable suggestions. We agree that the CO conversion largely depends on the olefin concentration when it is limited. So we increased the olefin concentration by ten times (2 mmol) at a fixed CO concentration (0.9 mmol). After reaction at 170 °C for 10 h, this system presented a CO utilization of up to 62.4% (**Supplementary Table 1 entry 3**). According to your suggestions, the CO utilization data have been added in **Fig. 1e and Fig. 1f** in the revised manuscript. Corresponding text has been added to the manuscript, attached here for your reference.

Revision: Page 3 Para. 2 Styrene conversion is optimal near ambient pressures of CO (0.06–0.1 MPa) and decreases rapidly with increasing pressure, so does the CO utilization, which is likely limited by styrene concentration. Then we increased the styrene concentration by tenfold at a fixed CO pressure, and achieved a CO utilization of 62.4% (Supplementary Table 1). After 16 hours of reaction the styrene conversion reached > 95% and the ester yield reached 72%, with the CO utilization of 23.7%.

Fig. 1e-f. The reaction evolutions with (e) CO pressure and (f) time over Ru/NbO_x. Reaction conditions: 2 wt.% Ru/NbO_x (30 mg), styrene (0.2 mmol), MeOH (2 mL), 170 °C, 10 h in (e), 0.1 MPa CO in (f).

2) The commendable performance of the Ru/NbO_x catalyst under comparatively lower CO partial pressures suggests distinctive reaction kinetics compared to comparative catalysts. To shed light on these differences, I recommend the experimental determination and reporting of the reaction orders for CO and methanol for both the newly developed solid catalyst and a relevant benchmark catalyst.

Response: Thanks for your good suggestions. We performed a series of kinetic experiments to determine the reaction orders of CO and styrene, and the results were shown in **Supplementary Fig. 1**. Based on the rate equation $v_b = kc_b^n$, wherein $\ln v_b = n \ln c_b + \ln k$, we estimated that the reaction order of styrene is 0.88, and the rate constant k is 2.9×10^{-5} at 170 °C. Within the investigated CO pressure range (0.1~1 bar), the reaction order of CO is 0.76, with the rate constant k of 5.6×10^{-10} . In addition, methanol serves as a significantly excessive solvent, keeping the nearly constant concentration throughout the reactions, which thus has almost no impact on the reaction rate in this study.

Supplementary Fig. 1 Reaction kinetics of methoxycarbonylation. (a) Reaction orders for styrene (a) and CO (b). Reaction conditions: (a) 20 mg Ru/NbO_x, styrene (0.01~0.1 mol/L), 2 mL MeOH, CO 0.1 MPa, 600 rpm, 170 °C for 2 h. (b) 20 mg Ru/NbO_x, styrene (0.1 mol/L), 2 mL MeOH, CO (10⁴ Pa ~10⁵ Pa), 600 rpm, 170 °C for 2 h.

Revision: Page 3 Para. 3 Kinetic experiments were conducted to establish apparent activation energy and the reaction orders of CO and styrene, as illustrated in Supplementary Fig. 1-3. The reaction order for styrene was determined as 0.88, with the corresponding rate constant k is 2.9×10^{-5} at 170 °C. Within the investigated CO pressure range (0.1~1 bar), the reaction order for CO is 0.76, with the rate constant k of 5.6×10^{-10} (Supplementary Fig. 1).

3) Anticipating a lower driving force for CO solubilization in the liquid reaction medium under lower CO partial pressures, I suggest a thorough assessment of the kinetic significance of mass transfer steps under the applied reaction conditions. This is important to corroborate that the rates determined experimentally faithfully correspond to the hydroxycarbonylation reaction. Experiment: the kinetic significance of mass transfer steps

Response: Thanks for your suggestion. The typical methoxycarbonylation reactions throughout this work run at a stirring rate of 600 rpm. To investigate the potential mass transfer effects, we examined the reaction performance at stirring rates of 400 rpm and 800 rpm. The corresponding reaction rates were depicted in **Supplementary Fig. 2**. Notably, the reaction rate was obviously lower at 400 rpm, highlighting the impact of mass transfer on the reaction. Fortunately, the reaction rate at 600 rpm closely resembled that at 800 rpm, suggesting that a stirring rate of 600 rpm is sufficient to eliminate potential mass transfer effects.

Supplementary Fig. 2 The effects of the stirring rate on the reaction rate of methoxycarbonylation. Reaction conditions: 20 mg catalyst, styrene (0.2 mmol), 2 mL MeOH, CO 0.1 MPa, 170 °C for 2 h.

Revision: Page 3 Para. 3 In a typical methoxycarbonylation reaction, the reaction was performed at a stirring rate of 600 rpm. Notably, the reaction rate was obviously low at 400

rpm, whereas the reaction rate at 600 rpm closely resembled that at 800 rpm, suggesting the elimination of potential mass transfer effects (Supplementary Fig. 2).

4) DFT methods: The description of the DFT methods lacks sufficient detail. Additional elaboration on the DFT methods employed in this study is necessary to provide a comprehensive description of the computational approach. The rationale behind selecting Ru(101) and Nb₂O₅ (001) facets for constructing the slab models in DFT calculations needs clarification too.

Response: Thanks for your suggestion. We have added the more detailed description of the DFT methods, including the weak interaction, free energy correction and so on. Specifically, Ru (101) and Nb₂O₅ (001) facets were selected for constructing the slab models in DFT calculations based on their facet stability. The choice of Nb₂O₅ (001) was justified by its presence as the most stable crystal facet in the XRD pattern (Supplementary Fig. 6) and its clear visibility in the HRTEM images (Fig 2a-2b). The *hcp* Ru (101) was selected for the crystal facet stability, known as the most stable facet for Ru, based on XRD diffraction patterns and literature references (*ACS Nano 2022, 16, 14885*).

Revision: Page 16 DFT Calculations The empirical correction in Grimme's scheme was used to describe the van der Waals interactions⁵³. Both Ru(101) and Nb₂O₅(001) surfaces were modelled by a four-layer slab model with a vacuum of 15 Å, which correspond to a k-points mesh of 2×2×1 and 2×2×1, respectively. The choice of Nb₂O₅ (001) was justified by its presence as the most stable crystal facet in the XRD pattern (Supplementary Fig. 6) and its clear visibility in the HRTEM images (Fig 2a-2b). The *hcp* Ru (101) was selected for the crystal facet stability, known as the most stable facet for Ru, based on XRD diffraction patterns and literature references.³⁸ During structural optimization, the bottom two layers of the surface were fixed at the bulk truncated position, and the top two layers and the adsorbates were fully relaxed. The adsorption energy ($E_{\text{ads}}(\text{X})$) of adsorbate X on the surface was calculated with the equation:

$$E_{\text{ads}}(\text{X}) = E_{\text{X_surf}} - E_{\text{surf}} - E_{\text{X}}$$

where E_{X} , E_{surf} and $E_{\text{X_surf}}$ are the total energies of the adsorbate X in gas phase, clean surface and the surface with adsorbate X, respectively. The negative $E_{\text{ads}}(\text{X})$ indicates the stronger adsorption strength. Noteworthy, the Gibbs free energy change (ΔG) and the barrier (E_{a}) of the elementary step were estimated ($\Delta G = \Delta H + \Delta E_{\text{ZPE}} - T\Delta S$), including the thermodynamic zero point energy (ZPE) correction and the entropy contribution ($T\Delta S$). Regarding $T\Delta S$ and ΔE_{ZPE} , we calculated the vibrational frequencies of the surface intermediates and transition states within DFT calculation;⁵⁶ for the gaseous molecules, the entropy contributions ($T\Delta S$) derived from the experimental values.

5) Further elaboration is needed on the differences in the slab models developed for O_v-NbO_x (partially reduced) and Nb₂O₅. It is difficult to perceive differences between the models as well as the optimized structures for the reaction transition states in the figures. Hence, I suggest to add all structural data of the calculations, including optimized xyz coordinates and

corresponding total energy, to the Supplementary Information.

Response: Thanks for your advice. The $\text{NbO}_x\text{-O}_v$ is the modified Nb_2O_5 with oxygen vacancies, which can be created through reduction treatment in hydrogen; thus in the simulation, the model of $\text{NbO}_x\text{-O}_v$ was constructed by the model of $\text{Nb}_2\text{O}_5(001)$ through creating oxygen vacancies. In addition, we have added the optimized xyz structural coordinates of the crucial intermediates and transition states, with the corresponding total energy, thermodynamic zero point energy (*ZPE*) correction and the entropy contribution (*TΔS*), attached in **Supplementary Note 7 (Page 25 to Page 112)**.

Revision: Page 13 Para. 1 The structural data of the calculations, including the optimized xyz structural coordinates of the crucial intermediates and transition states, the corresponding total energy, thermodynamic zero point energy (*ZPE*) correction and the entropy contribution (*TΔS*) are shown in Supplementary Note 7.

6) Presenting the predicted free energy diagram at the relevant reaction temperature would be more insightful than the E diagram. The reaction mechanism presented as most feasible has a reaction barrier of 1.15 eV. This appears to me a rather high barrier for a reaction which proceeds with relevant TOF at temperatures around 150°C. Can the authors compare this to an experimentally obtained apparent activation energy?

Response: Thanks for your good suggestions. In the revised manuscript, the Gibbs free energy changes (ΔG) and the barriers (E_a) of the elementary steps were estimated, including the thermodynamic zero point energy (*ZPE*) correction and the entropy contribution (*TΔS*), which have been updated in **Fig. 4 and Fig. 5** in the revised manuscript. The detailed DFT method has been added in **Methods DFT Calculations**.

Taking into account these contributions of *ZPE* and *TΔS*, the reaction barrier slightly decreases to 0.96 eV (i.e., 92.6 kJ/mol). In contrast, kinetic experiments were conducted to determine the apparent activation energy. The apparent activation energy for methoxycarbonylation reactions was determined as 60 ± 11 kJ/mol (**Supplementary Fig. 3**), basically aligning with the theoretical results.

Supplementary Fig. 3 The determination of apparent activation energy of methoxycarbonylation reactions. Reaction conditions: 20 mg catalyst, 0.2 mmol styrene, 2 mL MeOH, CO 0.1 MPa, 600 rpm, 150~180 °C for 2 h.

Revision: Page 2 Para. 3 The apparent activation energy of methoxycarbonylation reactions was determined as 60 ± 11 kJ/mol (Supplementary Fig. 3), basically aligning with the theoretical results (0.96 eV).³⁵

7) In situ FTIR spectroscopy: the discussion requires significant clarifications. Authors indicate in the text “then introducing a methanol/styrene solution into the cell”. However, the methods section suggests introduction of reactants in the vapor phase. This should be clarified. Please provide schemes of the cell system applied for these spectroscopic studied in the ESI.

Response: We thank you for your attention. It should be a methanol/styrene saturated vapor that was introduced into the in situ transmission cell. Apologize for the typographical error. For clarity, we carefully revised the corresponding discussion in the revised manuscript, illustrated the system studied and added **supplementary Fig. 15** in the revised Supporting Information, shown as follows.

Revision: Page 6 Para. 2 The scheme of the in situ FTIR system used in this work was presented as Supplementary Fig. 15.

Supplementary Fig. 15 The illustration scheme of in situ infrared spectroscopic system applied in this work.

Revision: Page 6 Para. 2 In situ Fourier-transform infrared (FTIR) surface reactions were subsequently conducted to experimentally evidence the methoxycarbonyl pathway. The illustration scheme of the in situ infrared spectroscopic system applied in this work was depicted in Supplementary Fig. 15. Temperature-resolved FTIR experiments were conducted by saturating the catalyst with CO at 30 °C, followed by evacuating the cell. Subsequently a methanol/styrene saturated vapor was introduced into the cell under vacuum until the system remained steady, and the infrared spectra were recorded. Stretching vibrational bands that are indicative of O–H, C=O, and C_{Ar}–C_{Ar} bonds in methanol, CO, and styrene are observed at 3700,

2080, and 1450 cm^{-1} , respectively (Fig. 6a). As the temperature jumps from 30 °C to 60 °C, a new band attributed to C=O stretching vibrations emerged at 1750 cm^{-1} , exhibiting an increasing intensity over the temperature, indicating the gradual formation of *COOMe species.^{43, 44} The stretching vibration frequency of C=O bond of *COOMe is further confirmed by the simulation using Vienna Ab initio Simulation Package, at 1720 cm^{-1} (Supplementary Fig. 16). The O–H vibrations of methanol at 3700 cm^{-1} decreased by nearly 50 % at 150 °C, due to both methanol consumption (for *COOMe formation) and heat-induced desorption. In contrast to the Ru/Nb₂O₅ analogue, the Ru/NbO_x catalyst showed a distinct peak at 1750 cm^{-1} , confirming its efficiency in generating *COOMe (Supplementary Fig. 10 and 11). The time-resolved FTIR experiments were conducted at 150 °C to further investigate the surface species. The procedures were similar to the above experiment except that the experimental temperature was fixed at 150 °C. Following CO adsorption and evacuation in vacuum, a methanol/styrene saturated vapor was introduced into the cell, and the infrared spectra were recorded. Immediately the vibrations at 1750 cm^{-1} appeared and increased (by orders of magnitude) with time (Fig. 6b).

8) The band assignment remains unclear and questionable, particularly with single IR bands being assigned to polycarbonyl ($\text{Ru}^{\delta+}(\text{CO})_n$) species. Addressing the discrepancy concerning the expected multiple C-O stretching vibrations (symmetric and asymmetric) for polycarbonyls is essential.

Response: We thanks for your carefulness on assignment. Based on the well-documented literature reported, we compiled a summary on C-O bond assignments of Ru carbonyl species on Ru/oxide catalysts, listed in **Supplementary Table 14** as follows. Tricarbonyl or dicarbonyl $\text{Ru}^{\delta+}(\text{CO})_n$ species typically presented two high frequency at ~2130 and ~2070 cm^{-1} , which have been reported on Ru/Nb₂O₅, Ru/TiO₂, Ru/SiO₂, Ru/Al₂O₃ catalysts (**Ref. 1-5**). Based on these reported results, we have cautiously assigned all the carbonyl species present in the revised **Fig. 2h** and **Fig. 6c-d**.

Revision: Page 6 Para. 3 In CO-probed FTIR spectra, the corresponding assignments of IR bands of carbonyl Ru species were assigned (Fig. 2h), based on the well-documented literatures (Supplementary Table 14).⁴⁵⁻⁴⁹ Tricarbonyl or dicarbonyl $\text{Ru}^{\delta+}(\text{CO})_n$ species typically present two high frequency at $\sim 2130 \pm 15$ and $\sim 2070 \pm 15$ cm^{-1} , the monocarbonyl $\text{Ru}^0(\text{CO})$ species shows a frequency at 2030 ± 30 cm^{-1} . Two in situ FTIR experiments were conducted to gain further insights into the active Ru sites of the catalyst (Fig. 6c-d).⁴⁰⁻⁴⁴

Supplementary Table 14 The reported band assignment of various Ru carbonyl species absorbed on oxide in CO-probed infrared spectra.

Adsorbed species	Ru adsorption sites	Frequency (cm^{-1})	Samples	Ref.
Ru(CO)₃	Ru ⁿ⁺ (n=1-3)	2142-2028,	Ru/Nb ₂ O ₅ , Ru/TiO ₂ ,	1-5
		2082-2067	Ru/SiO ₂ , Ru/Al ₂ O ₃	
	(SiO) _x Ru ⁿ⁺ (n=1-3)	2140, 2080	Ru/SiO ₂	2-3
	(RuO) _x Ru ⁿ⁺ (n=1-3)	2130, 2070		

Ru(CO)₂	Ru ²⁺	2132, 2070	Ru/SiO ₂	2
	Ru ⁰	2080, 2015	Ru/SiO ₂	2
Ru-CO	Ru ⁿ⁺ (n=1-3)	2110-2080	Ru/SiO ₂	2, 3
	Ru ⁰	2050-2010	Ru/Nb ₂ O ₅ , Ru/TiO ₂	1-5
	Ru ^{δ+}	1995	Ru/TiO ₂	2,4
	interfacial (TiO)Ru	1975	Ru/TiO ₂	4
		1950	Ru/TiO ₂	5
bridged Ru₂-CO	Ru ⁰	1990-1750	Ru/SiO ₂	3
	Ru ⁰	1810	Ru/TiO ₂	4

- [1] Komanoya, T.; Kinemura, T.; Kita, Y.; Kamata, K.; Hara, M. *J. Am. Chem. Soc.* 139 (2017) 11493-11499.
 [2] S.Y. Chin, C.T. Williams, M.D. Amiridis, *J. Phys. Chem. B* 110 (2006) 871-882.
 [3] G.H. Yokomizo, C. Louis, T. Bell, *J. Catal.* 120 (1989) 1-14.
 [4] Panagiotopoulou, P.; Kondarides, D. I.; Verykios, X. E. *J. Phys. Chem. C* 115 (2011) 1220-1230.
 [5] Zhou, J.; Gao, Z.; Xiang, G.; Zhai, T.; Liu, Z.; Zhao, W.; Liang, X.; Wang, L. *Nat. Commun.* 13 (2022) 327.

Fig. 2h CO-probe FTIR of 2 wt.% Ru/NbO_x catalyst.

9) The ascription of the C-O stretching vibration to Ru atoms interfaced with niobia requires further explanation. Why would the authors expect a higher degree of d-π* metal-to-CO backdonation for interfacial sites? I encourage the use of DFT methods to predict vibrational frequencies for relevant adsorbates and various sites to support band assignments.

Response: Thanks for your question. As summarized in the above **Supplementary Table 14**, the low-frequency $\sim 1960 \pm 15 \text{ cm}^{-1}$ of the carbonyl species on Ru/TiO₂, was considered as a contribution from interfacial (TiO)Ru-CO species, i.e., CO adsorbed on Ru sites located at the metal-support interface (*Nat. Commun.* 2022, 13, 327), shown as follows. It has been attributed to the strong electron-donating properties of noble metal atoms located at the metal-support

interface (e.g., M-Ti³⁺-O_v, or M-Nb-O_v sites), namely a stronger metal-to-CO backdonation, which originates from strong interaction with the reducible support (*Catal. Today* 181 (2012) 138–147). Bands at ~1960 cm⁻¹ of such interfacial (oxide)Ru(CO) species have been reported for a number of noble metals dispersed on reducible supports, including Pt/TiO₂ [*J. Catal.* 240 (2006) 114–125], RuTiO₂(Ca²⁺) [*J. Catal.* 198 (2001) 195–207], Au/TiO₂ [*J. Catal.* 188 (1999) 176–185], Pt/CeO₂ [*J. Catal.* 191 (2000) 30–45], [*Chem. Eng. J.* 134 (2007) 16–22] and Pd/CeO₂ [*J. Chem. Soc. Faraday Trans.* 92 (1996) 3233–3237]. Based on these reports, one can attribute 1970 cm⁻¹ to be interfacial (NbO)Ru(CO) species in Ru/NbO_x catalysts.

Fig.R1 a. Adsorption configurations of CO on Ru/TiO₂ catalysts. **b.** CO-DRIFTS of Ru/TiO₂ catalysts. (*Nat. Commun.* 2022, 13, 327).

10) The spectroscopic features associated with *COOMe species, crucial for supporting the mechanistic proposal, appear remarkably weak, fundamentally undiscernible from the spectral background. The adequacy of these features in conclusively proving the development of these reaction intermediates should be addressed.

Response: We fully agree with your viewpoint on the significance of the identification of *COOMe. As shown in **Fig. 6a-b**, the spectroscopic feature of *COOMe species at 1750 cm⁻¹ can be clearly observed by both in situ temperature-revolved and time-revolved infrared experiments. To make it clearer, we marked “COOMe” in red and bold in **Fig. 6a-b**.

As your suggest, the C=O bond stretching vibration frequency of *COOMe adsorbed on Ru is further simulated using the Vienna Ab initio Simulation Package. With the help of the Phonopy package, we conducted the frequency calculation in three directions and calculated the Born charge for the related structure. As shown in **Supplementary Fig. 16**, the stretching vibration frequency of C=O of *COOMe with strong oscillator strength is located at ~1720 cm⁻¹, which matches with the experimental results.

Fig. 6a-b Experimental evidence for the methoxycarbonyl species formation. In situ FTIR surface reactions of CO, styrene and methanol over Ru/NbO_x: (a) temperature-resolved reaction after chemisorption of CO and methanol/styrene vapor. (b) time-resolved process at 150 °C.

Supplementary Fig. 16 The simulated infrared vibration frequency of *COOMe using the Vienna Ab initio Simulation Package (VASP).

Revisions: Page 6 Para. 2 The C=O bond stretching vibration frequency of *COOMe is further confirmed by the simulation using Vienna Ab initio Simulation Package, observed at 1720 cm⁻¹ (Supplementary Fig. 16).

11) While the assessment of Ru dispersion on different catalysts based on CO chemisorption is noted, providing a rationale for the chemisorption stoichiometry factors considered in dispersion calculations is required. Additionally, X-ray absorption spectroscopy is suggested for an independent and more reliable assessment of the average metal dispersion.

Response: Based on the theoretically assumed stoichiometric ratio of 1:1 (metal : CO), Ru dispersion of 2 wt.% Ru/NbO_x was determined as 44.4%, and the corresponding Ru diameter

is estimated as 2.2 nm (**Supplementary Fig. 4** and **Supplementary Table 12**). Given the formation of polycarbonyl species [$\text{Ru}(\text{CO})_3$ and $\text{Ru}(\text{CO})_2$], the stoichiometry factors of metal-to-CO would be slightly lower than 1/1. Hence based on the CO-FTIR spectra (**Fig. 2h**) and the corresponding convolution results (**Supplementary Table 13**), we can estimate the stoichiometry factor of $n_{\text{Ru}}/n_{\text{CO}}$ is 0.822, according to the method previously reported (*Nat. Catal.* 2022, 5, 485–493). The calibrated dispersion of 2wt.% Ru/NbO_x catalyst is 36.4% (**Supplementary Table 13**).

X-ray absorption spectroscopy (XAS) characterizes the bulk phase and thereby provides overall average information. It is a powerful technique to investigate the structural and electronic properties of catalysts. The XANES and EXAFS of 2wt.% Ru/NbO_x catalyst were tested and added as **Fig. 3** in the revised manuscript. The best-fitted EXAFS result of Ru/NbO_x catalysts revealed Ru-O at 1.94 Å with coordination number (CN) of 3.6, Ru-Ru at 2.67 Å with CN of 5.1, and Ru-O-Nb/Ru shell at 3.08 Å with CN of 1.3, respectively (**Supplementary Table 3** and **Supplementary Fig. 7**). Given the first shell CN of 8.7 and *hcp* structure of Ru clusters, the size of clusters can be estimated as ~2.0 nm (pyramid or half-sphere shape), according to the reported literature (*Phys. Chem. Chem. Phys.* 2010, 12, 5562-5574). It showed a similar diameter and dispersion of Ru with CO chemisorption results.

Fig. 3 The electronic and coordinative structures of 2 wt.% Ru/NbO_x catalyst. (a) The normalized X-ray absorption near-edge spectra (XANES) at the Ru K-edge. (b) The k²-weighted Fourier transform extended X-ray absorption fine structure spectra (EXAFS) in r-space. (c) Wavelet transforms of Ru foil, Ru/NbO_x, and RuO₂.

Supplementary Table 3. The best-fitted EXAFS results of Ru/NbO_x^a.

Sample	Shell	CN	R (Å)	σ ² (10 ⁻² Å ²)	ΔE ₀ (eV)	r-factor (%)
Ru foil	Ru-Ru	12	2.67	-	-	-
	Ru-O	3.6±0.4	1.94±0.02	0.3±0.1	-7.0±1.8	
Ru/NbO _x	Ru-Ru	5.1±0.9	2.67±0.01	0.3±0.1	-7.0±1.8	0.9
	Ru-O-Nb/Ru	1.3±0.4	3.08±0.05	0.3±0.1	-7.0±1.8	

^aCN is the coordination number for the absorber-backscattered pair, R is the average absorber-backscattered distance, σ² is the Debye-Waller factor, and ΔE₀ is the inner potential correction. * S₀² was fixed to 0.72 as determined from Ru foil fitting. The data range used for data fitting in k-space (Δk) and R-space (ΔR) are 3.0-12.7 Å⁻¹ and 1.0-3.3 Å, respectively.

Supplementary Fig. 7 EXAFS fitting of Ru/NbO_x catalyst. (a) Ru K-edge EXAFS fitting curve, shown in k² weighted k-space. (b) EXAFS fitting curve in the region of 1.0-3.3 Å, shown in k² weighted R-space.

12) Notation: referring to "solid" and "molecular" catalysts rather than "heterogeneous" and "homogeneous" catalysts, respectively, is proposed. The latter adjectives are more appropriately applied to "catalysis" than to the "catalysts" involved.

Response: Thanks for your advice. We revised the corresponding terms into "solid" and "molecular" catalysts in **Fig. 1b** and the text in the manuscript.

Reviewer #2 (Remarks to the Author):

The manuscript “Ambient-pressure Alkoxy carbonylation for Sustainable Synthesis of Ester” demonstrates the use of Ru supported over NbO_x for alkoxy carbonylation of olefins at ambient pressure of CO. This work is interesting in terms of studying the mechanism of the reaction using in-situ FTIR and DFT. However, I am not convinced regarding its higher efficiency compared to earlier published heterogeneous catalysts. Firstly, the term CO utilization, to which the authors refer, does not seem very important for comparison. It increases at low pressure but also depends on the volume of the reactor. Thus, it is possible to significantly increase it by using a small volume reactor. The earlier published Ru/CeO₂ catalyst (Chinese Journal of Catalysis 41 (2020) 963–969, J. Am. Chem. Soc. 2018, 140, 11, 4172–4181) also demonstrates high activity and selectivity under similar conditions and shows an increase in activity at atmospheric pressure. Therefore, I would recommend comparing the performance with this catalyst under the same conditions in the same reactor. This would provide a clearer vision about the advantage of NbO_x as a support. Although there are some mechanistic studies, I wouldn't say that this work is sufficiently new to be published in Nat Com. I think it would be better suited to a more specialized journal. Additionally:

Response: Thanks for your comments. The CO utilization (rate) is defined as the molar ratio of the yielded esters to the total CO inputs. As shown in the **equation (1)**, the key to increasing CO utilization is to enhance the contribution of CO inputs to ester products. Hence, if the catalysts have the similar carbonylation efficiency, using a small volume reactor with a decreased alkene amount (a constant ratio of $n_{\text{styrene}}/V_{\text{reactor}}$) would present a similar CO utilization rate. In the earlier published work, 0.5 mmol styrene in 50 mL reactor was adopted (0.01 mol_{styrene}/L_{reactor}). Herein we adopted a similar $n_{\text{styrene}}/V_{\text{reactor}}$ ratio (0.011 mol_{styrene}/L_{reactor}, 0.188 mmol in 16 mL). Therefore, the reaction conditions for evaluating CO utilization rate and carbonylation performance are comparable. Based on a similar condition, we calculated the TOF (h⁻¹) of carbonylation for Ru/NbO_x catalysts, which is higher than the Ru/CeO₂ reported earlier, as shown in **Table R1** (More details in **Supplementary Table 1**). It demonstrates the advantages of Ru/NbO_x catalysts.

$$\text{CO utilization} = \frac{n_{\text{ester}}}{n_{\text{CO}}} \times 100\% = \frac{n_{\text{ester}}}{pV_{\text{CO}}/RT} \times 100\% \quad (1)$$

Table R1. Comparison of CO utilization and carbonylation performance of state-of-the-art catalysts for alkoxy carbonylation of alkenes.

Entry	Catalyst	Substrate	n_{Sub} (mmol)	V_{CO} (ml)	Conditions			Yield (%)	TOF (h ⁻¹)	CO utilization (%)	CO utilization efficiency (mmol _{COO} mol _{CO} ⁻¹ h ⁻¹)	Reference
					CO (bar)	T/°C	t/h					
1	Ru/NbO _x	styrene	0.188	14	1.6	170	16	71	4.1	14.8	9.2	this work
2			1.88	14	1.6	170	10	60	8.4	62.4	62.4	
3		ethylene	0.57	14	1	170	1	33.6	33.8	33.6	336	
4			1.13	14	1	170	2	22.5	22.7	45.0	225	
5	Ru/CeO ₂	styrene	0.5	46	2	165	14	80	1.4	10.8	7.7	Chin. J. Catal. , 2020, 41, 963-969.
6		ethylene	7.5	46	7	165	8	62	19.6	35.8	44.7	J. Am. Chem. Soc. , 2018, 140, 4172-4181.

The earlier two publications reported the initial cases in heterogeneous methoxycarbonylation and obtained linear-dominated esters (*Chin. J. Catal.* 2020, 41, 963-969. *J. Am. Chem. Soc.* 2018, 140, 11, 4172–4181). The work is innovative and highlights that this field is in its earliest infancy, with vast important unknowns yet to be explored. In this work, we not only developed an efficient catalyst but also made an in-depth investigation on the reaction pathway and mechanism, by combining in situ infrared experimentations and DFT calculations. To make the novelty of this work more clear, we made a significant revision in the **Abstract** and **Introduction**.

1. I recommend performing a comparison with already published O-deficient supports, such as CeO₂, to demonstrate the efficiency of Ru/NbO_x. I also couldn't find activity over Ru/Nb₂O₅ and reference samples in Fig. 2.

Response: Thank you for your valuable suggestions. As suggested, we prepared Ru/CeO₂ and O-deficient Ru/CeO_x catalysts according to the methods described (*J. Am. Chem. Soc.* 2018, 140, 11, 4172–4181), and evaluated their catalytic performance in the same reactor. Under the identical condition (160 °C, 10 h, 1 MPa CO), Ru/NbO_x catalysts presented a two-fold increase in carbonylation yield (12%) compared to Ru/CeO_x (5%). These results are presented in **Supplementary Table 2** (Entries 8 and 11). As you suggested, the catalytic performance of Ru/Nb₂O₅ has been added into **Fig. 1d**. To enhance clarity, the color of the conversion dot for Ru/NbO_x has been changed to white, as follows in **Fig 1d**.

Fig. 1d. Catalytic methoxycarbonylation reaction. (d) Screening solid catalysts. Reaction conditions: 5 wt.% M/C (20 mg) or 2 wt.% Ru/oxide (30 mg), styrene (0.2 mmol), MeOH (2 mL), 160 °C, 10 h, CO (1 MPa) or ★ (0.1 MPa). Other is primarily the dimer of styrene.

2. There are no results regarding the stability of the catalyst for several cycles. Most probably, O vacancies are deactivated in the presence of methanol, leading to a decrease in activity over time.

Response: Thanks for your valuable suggestions. We conducted a cycling experiment using both ethylene and styrene substrates. The reused catalyst showed a decreased carbonylation yield. The Ru 3d XPS, ICP-OES, and EPR characterizations were performed for the cycled

catalysts, revealing that the deactivation was attributed to an obvious decline in oxygen vacancies (**Supplementary Fig. 20**). The calcination treatments for the cycled catalysts were performed in both air and H₂, yet it remained difficult to recover the carbonylation activity. It is a complicated and challenging issue, which we aim to address to improve the recyclability in the near future. The corresponding table and figure were added as **Supplementary Table 21 and Supplementary Fig. 20**.

Supplementary Table 21. Cycle performance.

Entry	Substrate	Cycle	Conversion (%)	Yield of carbonylation(%)
1		1	31.7	29.4
2	ethylene ^a	2	3	3
3		3	4	4
4	styrene ^b	1	70	52
5		2	7	2

Reaction conditions: ^a2%Ru/NbO_x (30 mg), ethylene (0.57 mmol), MeOH (2.0 mL), CO (1 bar), 170 °C, 2 h. ^b2%Ru/NbO_x (30 mg), styrene (0.2 mmol), MeOH (2.0 mL), CO (1 bar), 170 °C, 10 h.

Supplementary Fig. 20 Characterizations of Ru/NbO_x catalysts before and after carbonylation reaction. (a) EPR and (b) Ru 3d XPS of Ru/NbO_x before and after reaction.

Revisions: Page 15 Para. 1 The cycling experiments and the corresponding characterizations of the cycled catalysts are shown in Supplementary Table 21 and Supplementary Fig.20.

3. The mechanism proposed by the authors involves the conversion of Ruⁿ⁺(CO)_n species to metallic Ru with the formation of *COOMe intermediate, indicating the synergetic role of Ruⁿ⁺ and Ru metallic. However, this would mean that the reaction is not catalytic, or the mechanism of regeneration of cationic Ru should be proposed. It seems unclear.

Response: We appreciate your great advice. The mechanism didn't involve the transformation of ionic Ru to metallic Ru, but rather the release of CO from polycarbonyl Ru^{δ+}(CO)_n species. As shown in **Fig. 6c**, the C-O vibration intensity of Ru^{δ+}(CO)_n weakened immediately upon

introducing styrene. The signal of adsorbed styrene increased, but the intensity of metallic $\text{Ru}^0(\text{CO})$ remained almost constant. This demonstrates that $\text{Ru}^{\delta+}(\text{CO})_n$ released partial CO over $\text{Ru}^{\delta+}$ sites, followed by styrene adsorption, due to the stronger adsorption of styrene. To confirm this, we calculated the adsorption energy of styrene over Ru surface with high CO coverage (-0.65 eV), which was -3.27 eV for styrene, indicating the stronger adsorption of styrene. To further experimentally confirm this, we conducted an infrared experiment where styrene first adsorbed to saturation over Ru/NbO_x catalyst, followed by CO adsorption. As shown in **Fig. R4**, no CO adsorption was observed after the saturated adsorption of styrene, indicating that styrene adsorbed more strongly than CO on Ru catalysts. Additionally, Ru/NbO_x catalysts, both before and after the reaction, contain both $\text{Ru}^{\delta+}$ and Ru^0 components, with a relatively unchanged proportion (Ru XPS spectra in **Supplementary Fig 20**). It indicates that $\text{Ru}^{\delta+}$ sites are responsible for CO enrichment and alkene adsorption. We apologize for any misunderstanding previously described. For clarity, the corresponding discussion was carefully revised in the revised manuscript, as follows.

Fig. 6c. In situ infrared spectra. Adsorption of styrene vapor after CO desorption at 150 °C.

Fig. R4. IR spectra of styrene adsorption followed by CO adsorption over Ru/NbO_x catalyst at room temperature. The styrene vapor was introduced into the cell under vacuum until the system remained steady, followed by CO adsorption until saturation and then sweeping by Ar.

Revision: Page 6 Para. 3 Ru 3d XPS results showed that Ru/NbO_x catalysts, both before and after the reaction, contained both Ru^{δ+} and Ru⁰ components, with a relatively unchanged proportion (Supplementary Fig 20). Based on these findings, we consider that metallic Ru⁰ and ionic Ruⁿ⁺ sites synergistically contribute to the reaction. This was strongly supported by a significant fall in the carbonylation yield over the NbO_x supported sole Ru⁰ catalysts (Supplementary Table 15 and Fig. 2g).

4. Nb oxide is the most acidic oxide support. What is the role of these acid sites in the reaction? Is it possible that these sites could stabilize Ru cationic? Or they are involved in adsorption of ethylene according to the mechanism proposed earlier for this reaction.

Response: Thank you for your questions. The acidic sites of Ru/NbO_x catalyst were identified by pyridine-probe infrared spectra (**Supplementary Fig. 12**). The acidic NbO_x support can facilitate the adsorption of alkenes, as evidenced by in situ infrared spectra (**Supplementary Fig. 9a**). The interfacial Ru-O-Nb-O_v structures (confirmed by EXAFS), consisting of Lewis acid-base pair sites, act as active sites for methanol dissociation (**Supplementary Fig. 9**) and hydrogen transfer for the final hydrogenation desorption (**Fig.4b**). The Ru^{δ+} species itself acts as a Lewis-acid, most likely stabilized by adjacent O or O_v (Lewis base) in NbO_x support.

Supplementary Fig. 12a. Pyridine-probe FTIR spectra of Ru/NbO_x catalyst. Pyridine desorption at 40 °C, 200 °C and 350 °C, respectively.

Supplementary Fig. 9 In situ FTIR on NbO_x support.

Revision: Page 5 Para. 1 The acidic sites of Ru/NbO_x catalyst were identified by pyridine-probe infrared spectra (Supplementary Fig. 12). The acidic NbO_x support can facilitate the adsorption of alkenes, as evidenced by in situ infrared spectra (Supplementary Fig. 9).

5. Earlier it has been found that Ru/CeO₂ contains Ru-O-Ce sites (*J. Am. Chem. Soc.* 2018, 140, 11, 4172–4181), what's about Ru-O-Nb in this catalyst?

Response: As suggested, we conducted XANES and EXAFS characterizations for Ru/NbO_x catalysts, added as **Fig. 3** in the revised manuscript (as follows). The Fourier-transformed k^2 weighted EXAFS spectra at the Ru K-edge and wavelet transform (WT) showed that the Ru catalyst presented three major peaks, ascribed to Ru-O, Ru-Ru, and Ru-O-Nb/Ru coordination, in contrast to the reference samples of Ru foil and RuO₂ (**Fig. 3b-3c**). Consistently, the best-fitted EXAFS result of Ru/NbO_x catalyst revealed Ru-O-Nb/Ru shell at 3.08 Å with coordination number (CN) of 1.3, respectively (**Supplementary Table 3**). The interfacial Ru-O-Nb-O_v sites consist of Ru sites and intimate O or O_v sites on the support (*J. Am. Chem. Soc.* 2018, 140, 11, 4172–4181). These sites are synergistically active for the adsorptive dissociation of CH₃OH on O_v sites (**Fig. 4a**), CO adsorption exclusively on Ru sites (**Supplementary Fig. 9 and 11**), and alkene adsorption on both Ru and acidic Nb-O_v sites (**Fig. 4-5, Supplementary Fig. 9 and 11**), therefore promoting the subsequent hydromethoxycarbonylation reaction of three substrates.

Fig. 3. The electronic and coordinative structures of Ru/NbO_x catalyst. (a) The normalized X-ray absorption near-edge spectra (XANES) at the Ru K-edge. (b) The k^2 -weighted Fourier transform extended X-ray absorption fine structure spectra (EXAFS) in r -space. (c) Wavelet transforms of Ru foil, Ru/NbO_x, and RuO₂.

Supplementary Table 3. The best-fitted EXAFS results of Ru/NbO_x^a.

Sample	Shell	CN	R (Å)	σ^2 (10^{-2} Å ²)	ΔE_0 (eV)	r-factor (%)
Ru foil	Ru-Ru	12	2.67	-	-	-
	Ru-O	3.6±0.4	1.94±0.02	0.3±0.1	-7.0±1.8	
Ru/NbO _x	Ru-Ru	5.1±0.9	2.67±0.01	0.3±0.1	-7.0±1.8	0.9
	Ru-O-Nb/Ru	1.3±0.4	3.08±0.05	0.3±0.1	-7.0±1.8	

^aCN is the coordination number for the absorber-backscattered pair, R is the average absorber-backscattered distance, σ^2 is the Debye-Waller factor, and ΔE_0 is the inner potential correction. * S_0^2 was fixed to 0.72 as determined from Ru foil fitting. The data range used for data fitting in k -space (Δk) and R -space (ΔR) are 3.0-12.7 Å⁻¹ and 1.0-3.3 Å, respectively.

Revision: Page 4 Para. 3 The electronic and coordinative structures of Ru/NbO_x catalysts are further characterised by X-ray absorption spectra (XAS) technique.³⁸⁻³⁹ Fig. 3a displays the X-ray absorption near-edge spectra (XANES) at Ru K-edge of Ru catalyst and references. The adsorption threshold E₀ for Ru/NbO_x is higher than Ru foil but lower than RuCl₃, suggesting most likely coexistence of metallic and ionic Ru speciation, consistent with the XPS characterization results. Further, the coordination environment of Ru atoms is determined by the extended X-ray absorption fine structure spectra (EXAFS). As shown in the Fourier-transformed k² weighted EXAFS spectra at the Ru K-edge (Fig. 3b), in contrast to the reference samples of Ru foil and RuO₂, the Ru catalyst showed three major peaks, which could be ascribed to Ru-O, Ru-Ru, and Ru-O-Nb/Ru coordination, respectively. To further resolve Ru-O, Ru-Ru, and Ru-O-Nb/Ru coordination, wavelet transform (WT) of Ru K-edge EXAFS oscillations was carried out owing to its more powerful resolutions in both *k* and *r* spaces (Fig. 3c).³⁸ The three hills at (1.45 Å, 5.00 Å⁻¹), (2.38 Å, 8.50 Å⁻¹), and (2.80 Å, 9.50 Å⁻¹), associated with Ru-O, Ru-Ru, and Ru-O-Nb/Ru contribution, respectively, is clearly observed from the WT contour plots of Ru/NbO_x catalyst. Consistently, the best-fitted EXAFS result revealed Ru-O at 1.94 Å with coordination number (CN) of 3.6, Ru-Ru at 2.67 Å with coordination number (CN) of 5.1, and Ru-O-Nb/Ru shell at 3.08 Å with coordination number (CN) of 1.3, respectively, in Ru/NbO_x (Supplementary Table 3 and Supplementary Fig. 7). The interfacial Ru-O-Nb-O_v sites consist of Ru sites and intimate O or O_v sites on the support. These sites are synergistically active for the adsorptive dissociation of CH₃OH on O_v sites (Fig. 4a), CO adsorption exclusively on Ru sites (Supplementary Fig. 9 and 11), and alkene adsorption on both Ru and acidic Nb-O_v sites (Fig. 4-5, Supplementary Fig. 9 and 11), therefore promoting the subsequent hydromethoxycarbonylation reaction of three substrates.

6. It would be important to provide more information about the structure sensitivity of the catalyst. What should be the size of Ru clusters for optimal performance? How does activity change depending on the size of the clusters?

Response: We thank you for the good suggestion. The optimal 2 wt.% Ru/NbO_x catalyst presented a active diameter of 2.2 nm, evaluated from CO pulse chemisorption. We compared the catalytic performance of the fully dispersed Ru₁/NbO_x catalyst (≤ 1.1 nm), 2 wt.% Ru/NbO_x (ca. 2.2 nm), and control Ru/NbO_x-3.1 nm (ca. 3.1 nm), as shown in **Supplementary Table 11-12 and Supplementary Fig. 4**. Notably, the carbonylation yield presented a volcano-curve tendency with the increased size of Ru clusters. The 2 wt.%Ru/NbO_x catalyst, featuring a diameter of 2.2 nm, exhibited a better activity (80%) and carbonylation yield (62%) compared to the others tested. The fully dispersed Ru₁/NbO_x showed only <1% conversion and no ester products were detected, most likely due to the absence of the accommodating Ru⁰ sites for multiple reactants (CO, MeOH and styrene). The Ru/NbO_x-3.1 nm exhibited a low carbonylation yield of 14% as well as a higher hydrogenation yield of 28%. The larger-size Ru nanoparticles favoured the side-reaction hydrogenation of alkene substrates, due to the enhanced metallicity.

Supplementary Table 11 Effects of fully dispersed Ru and size-dependent Ru clusters on styrene carbonylation.

Entry	catalysts	Conv. (%)	Yield (%)			L/(L+B) (%)	C. B. (%)
			carbonylation	hydrogenation	methoxylation		
1	Ru ₁ /NbO _x	0.8	0	0.8	0	-	90
2	Ru/NbO _x -2.2 nm	80	62	17	0	75	97
3	Ru/NbO _x -3.1 nm	42	14	27	0	74	95

Supplementary Table 12 Ru dispersion, CO pulse chemisorption data and ICP-OES data of Ru/NbO_x catalysts.

Sample	Ru wt.% ^a	CO accumulative sorption quantity (mmol/g) ^b	Ru dispersion ^b	Ru active diameter (nm) ^b
Ru ₁ /NbO _x	0.098%	0.00969	99.8%	1.1
2 wt.% Ru/NbO _x	1.891%	0.08172	44.4%	2.2
Ru/NbO _x -3.1 nm	3.331%	0.14432	43.8%	3.1

^aThe ruthenium loading was determined by inductively coupled plasma optical emission spectroscopy.

^bThe metal dispersion of catalysts was measured by CO pulse chemisorption.

Revision: Page 3 Para. 4 Next, the effect of the size-dependent Ru speciation was investigated. The fully dispersed Ru catalyst (Ru₁/NbO_x) and Ru/NbO_x catalyst with a diameter of 3.1 nm (Ru/NbO_x-3.1 nm) were synthesized (Supplementary Fig. 4 and Supplementary Table 12). Evaluated by the methoxycarbonylation of styrene, the activity of carbonylation presented a volcano-curve tendency with the size of Ru clusters (or nanoparticles). The 2 wt.% Ru/NbO_x catalyst, featuring a diameter of 2.2 nm, exhibited a better activity (80%) and carbonylation yield (62%), compared to the others tested (Supplementary Table 11). The fully exposed Ru₁/NbO_x showed that only <1% yield of hydrogenation was observed and no ester products were detected, most likely due to the absence of the accommodating reaction sites for multiple reactants. The Ru/NbO_x-3.1 nm exhibited a low carbonylation yield of 14% but a higher hydrogenation yield of 28%. The larger-size Ru nanoparticles favoured the side-reaction hydrogenation of alkene substrates, most likely due to the enhanced metallicity.

REVIEWER COMMENTS

Reviewer #1 (Remarks to the Author):

The authors have thoroughly revised their manuscript in response to the reviewers' comments. The new data and discussions significantly clarify and strengthen the conclusions. I recommend the revised manuscript for publication in Nature Communications.

Reviewer #2 (Remarks to the Author):

The authors did a great job addressing the reviewers' comments. My main concern is related to the stability test. The catalyst almost completely deactivates after the first cycle. This is a very serious issue because it prevents the catalyst use. The authors claim an obvious decline in oxygen vacancies, however, according to EPR analysis before and after the reaction, this decline is actually very small. It is also strange that the catalyst cannot be regenerated by calcination in air or hydrogen. I would suggest a deeper analysis of what happens to the catalyst. It could be that Ru agglomerates or redistributes after the reaction, considering the strong structure sensitivity effect, or it may transform into carbonyl. I recommend performing additional TEM analysis and FTIR analysis of CO and Py adsorption.

Other comments:

1. The revision of the text is insufficient. For example, there is no discussion concerning the use of Ru/CeO₂, although it shows similar performance to NbO_x-based catalysts.
2. The role of acid sites is still unclear. I suggest performing the reaction in the presence of a small amount of Py as an in-situ poison to clarify if it affects the catalytic performance.

REVIEWER COMMENTS

Reviewer #1 (Remarks to the Author):

The authors have thoroughly revised their manuscript in response to the reviewers' comments. The new data and discussions significantly clarify and strengthen the conclusions. **I recommend the revised manuscript for publication in Nature Communications.**

Response: Thank you very much for your positive feedback.

Reviewer #2 (Remarks to the Author):

The authors did a great job addressing the reviewers' comments. My main concern is related to the stability test. The catalyst almost completely deactivates after the first cycle. This is a very serious issue because it prevents the catalyst use. The authors claim an obvious decline in oxygen vacancies, however, according to EPR analysis before and after the reaction, this decline is actually very small. It is also strange that the catalyst cannot be regenerated by calcination in air or hydrogen. I would suggest a deeper analysis of what happens to the catalyst. It could be that Ru agglomerates or redistributes after the reaction, considering the strong structure sensitivity effect, or it may transform into carbonyl. I recommend performing additional TEM analysis and FTIR analysis of CO and Py adsorption.

Response: Thank you for the valuable suggestions. Following your comments, we conducted pyridine-probe and CO-probe FTIR, as well as aberration-corrected high-angle annular dark field scanning transmission electron microscopy (AC-HAADF-STEM) and EPR analysis on both fresh and used Ru/NbO_x catalysts from the same batch.

As shown in **Supplementary Fig. 20a**, pyridine-probe FTIR revealed a significant decline (ca. 50% or more) in Brönsted acid sites for the used catalyst compared to the fresh one, while the amount of Lewis acid sites remained nearly unchanged. CO-probe FTIR indicated a substantial decrease in exposed Ru species for the used catalyst, suggesting a potential structural change of Ru surface after the reaction (**Supplementary Fig. 20b**). The AC-HAADF-STEM images of the fresh catalyst depicted uniformly dispersed small Ru clusters over the NbO_x support (**Figure 3d-e**). In contrast, the used catalyst exhibited prevalent particulate Ru aggregates with no highly dispersed Ru clusters observed in the investigated regions (**Supplementary Fig. 21**). ICP-OES results also showed the loading of Ru metal slightly declined from 1.89% to 1.28%. Furthermore, EPR analysis of fresh and used catalysts was carefully remeasured using two independent instruments. Both results confirmed that the amount of oxygen vacancies in the used catalysts did not decline after reaction (**Supplementary Fig. 20c**). In addition, Ru 3d XPS results showed that Ru/NbO_x catalyst, both before and after the reaction, contained both Ru^{δ+} and Ru⁰ components with a slightly changed proportion (**Supplementary Fig. 20d**). Based on these analysis, we can conclude that the agglomeration of highly dispersed Ru clusters over support is most likely responsible for the observed deactivation of the catalyst.

Supplementary Figure 20 Characterizations of Ru/NbO_x catalysts before and after carbonylation reaction. (a) Pyridine-probe FTIR. (b) CO-probe FTIR. (c) EPR. (d) Ru 3d XPS.

Figure 3d-e AC-HAADF STEM images and the corresponding Ru and Nb element map of the fresh 2 wt.% Ru/NbO_x.

Supplementary Figure 21 AC-HAADF STEM images and the corresponding Ru and Nb element map of the used 2 wt.% Ru/NbO_x.

Revision: Page 3 Para 3 Despite the advantages of heterogeneous Ru/NbO_x catalyst, there remains an issue of cycling stability (Supplementary Table 21). Comprehensive characterization of the fresh and cycled catalysts from the same batch were conducted (Supplementary Fig. 20). Pyridine-probe FTIR revealed a significant decline (ca. 50% or more) in Brønsted acid sites for the used catalyst compared to the fresh one, while the amount of Lewis acid sites remained nearly unchanged. CO-probe FTIR indicated a substantial decrease in exposed Ru species for the used catalyst, suggesting a potential structural change of Ru surface during the reaction (Supplementary Fig. 20b). The AC-HAADF-STEM images of the used catalyst exhibited prevalent particulate Ru aggregates, while no highly dispersed Ru clusters were observed in the investigated regions (Supplementary Fig. 21). ICP-OES results also showed the loading of Ru metal slightly declined from 1.89% to 1.28%. Furthermore, EPR analysis of fresh and used catalysts was carefully measured and confirmed that the amount of oxygen vacancies in the used catalysts did not decline after the reaction (Supplementary Fig. 20c). Ru 3d XPS results showed that Ru/NbO_x catalyst, both before and after the reaction, contained both Ru^{δ+} and Ru⁰ components with a slightly changed proportion (Supplementary Fig. 20d). Based on these analyses, we can conclude that the agglomeration of highly dispersed Ru clusters over support is most likely responsible for the observed deactivation. Given the significance of this issue, we aim to address it to improve recyclability in the near future.

Other comments:

1. The revision of the text is insufficient. For example, there is no discussion concerning the use of Ru/CeO₂, although it shows similar performance to NbO_x-based catalysts.

Response: Thank you for the good suggestion. CeO₂-based catalysts possess high oxygen storage and release capacity, a key feature of CeO₂ that allows the formation of abundant oxygen vacancy (O_v) defects on its surface. These oxygen defects can act as active sites for anchoring and dispersing the active metal species (e.g. Ru), as well as methanol dissociation at the Ce-V_o site (*J. Am. Chem. Soc.* 2018, 140, 11, 4172–4181). Our recent several works found that NbO_x-based catalysts are effective at activating and dissociating the C-O bonds (*Nat. Commun.* 2021, 12(1): 9; *Chem. Eng. J.* 2024, 479, 147687; *CCS Chem* 2024, 6(3): 709-718). Although Nb₂O₅ is typically less reducible than CeO₂, its surface can create a large quantity of oxygen defects through high-temperature hydrogen treatment. The defective surface and its intrinsic acidic properties, make it potentially catalytically active for promoting hydromethoxycarbonylation reaction.

In addition, other several discussions were added into the manuscript.

Revision: Page 3 Para 1 The ceria-based Ru catalysts (Ru/CeO₂, Ru/CeO_x) showed distinct carbonylation selectivity (80% ~ 90%) despite poor conversion, due to their ability to form oxygen vacancy defects (O_v) on the surface. These oxygen defects can serve as active sites for anchoring and dispersing the Ru metal, as well as for methanol dissociation at the Ce-O_v site.^{34, 35} Our recent work has found that NbO_x-based catalysts are effective at activating and dissociating the C-O bonds,³⁸⁻⁴⁰ due to the abundant oxygen defects created through hydrogen treatment, despite Nb₂O₅ being typically less reducible than CeO₂. The defective surface, along with its intrinsic acidic properties, makes NbO_x-based catalysts potentially catalytically active for promoting the hydromethoxycarbonylation reaction.

Page 4 Para 2 To gain a more direct visualization of the metal distribution of 2 wt.% Ru/NbO_x, the aberration-corrected high-angle annular dark field scanning transmission electron microscopy (AC-HAADF-STEM) was performed. The images clearly demonstrated that ultrafine Ru nanoclusters were uniformly dispersed over the support (Fig. 3d-h). No large nanoparticles were observed in the regions investigated.

Page 5 Para 2 The field of heterogeneous methoxycarbonylation is in its earliest infancy, with vast important unknowns yet to be explored. Besides developing an efficient catalyst, understanding the reaction pathway and mechanism is of high significance.

2. The role of acid sites is still unclear. I suggest performing the reaction in the presence of a small amount of Py as an in-situ poison to clarify if it affects the catalytic performance.

Response: Following your suggestion, we performed the reaction with pyridine as a poison, as shown in **Supplementary Table 22**. The results demonstrate that pyridine as an in-situ poison did not affect the catalytic performance. The phenomenon aligns well with our previous regioselectivity regulation experiments (**Supplementary Figure 18**), where the addition of a small amount of acid (CH₃COOH, H₃BO₃) resulted in similar catalytic performance compared to no addition.

Supplementary Table 22 Effects of pyridine poison on styrene carbonylation.

Entry	Poison	Conv. (%)	Yield. (%)		
			carbonylation	hydrogenation	methoxylation
1	pyridine	74	49	16	0
2	no	73	50	14	0

Reaction conditions: 2 wt.%Ru/NbO_x (30 mg), styrene (0.2 mmol), pyridine (6.3 μmol), MeOH (2.0 mL), CO (1 bar), 170 °C, 10 h. The amount of pyridine added was 1.8 molar equivalent of the total acid sites (117 μmol) in the catalyst, based on the quantification value of pyridine-FTIR.

REVIEWERS' COMMENTS

Reviewer #2 (Remarks to the Author):

The authors clarified the role of Ru agglomeration in catalyst deactivation, the impact of acidity, and the reasons behind the similar performance of Ru/CeO₂. I expect the manuscript can be accepted for publication.